# An Accelerated Decentralized Stochastic Proximal Algorithm for Finite Sums

**Hadrien Hendrikx**
INRIA - DIENS - PSL Research University
`hadrien.hendrikx@inria.fr`

**Francis Bach**
INRIA - DIENS - PSL Research University
`francis.bach@inria.fr`

**Laurent Massoulié**
INRIA - DIENS - PSL Research University
`laurent.massoulie@inria.fr`

## Abstract

Modern large-scale finite-sum optimization relies on two key aspects: distribution and stochastic updates. For smooth and strongly convex problems, existing decentralized algorithms are slower than modern accelerated variance-reduced stochastic algorithms when run on a single machine, and are therefore not efficient. Centralized algorithms are fast, but their scaling is limited by global aggregation steps that result in communication bottlenecks. In this work, we propose an efficient **A**ccelerated **D**ecentralized stochastic algorithm for **F**inite **S**ums named ADFS, which uses local stochastic proximal updates and randomized pairwise communications between nodes. On $n$ machines, ADFS learns from $nm$ samples in the same time it takes optimal algorithms to learn from $m$ samples on one machine. This scaling holds until a critical network size is reached, which depends on communication delays, on the number of samples $m$, and on the network topology. We provide a theoretical analysis based on a novel augmented graph approach combined with a precise evaluation of synchronization times and an extension of the accelerated proximal coordinate gradient algorithm to arbitrary sampling. We illustrate the improvement of ADFS over state-of-the-art decentralized approaches with experiments.

## 1 Introduction

The success of machine learning models is mainly due to their capacity to train on huge amounts of data. Distributed systems can be used to process more data than one computer can store or to increase the pace at which models are trained by splitting the work among many computing nodes. In this work, we focus on problems of the form:

$$\min_{\theta \in \mathbb{R}^d} \sum_{i=1}^{n} f_i(\theta), \quad \text{where} \quad f_i(\theta) = \sum_{j=1}^{m} f_{i,j}(\theta) + \frac{\sigma_i}{2}\|\theta\|^2. \tag{1}$$

This is the typical $\ell_2$-regularized empirical risk minimization problem with $n$ computing nodes that have $m$ local training examples each. The function $f_{i,j}$ represents the loss function for the $j$-th training example of node $i$ and is assumed to be convex and $L_{i,j}$-smooth [Nesterov, 2013, Bubeck, 2015]. These problems are usually solved by first-order methods, and the basic distributed algorithms compute gradients in parallel over several machines [Nedic and Ozdaglar, 2009]. Another way to speed up training is to use *stochastic* algorithms [Bottou, 2010, Defazio et al., 2014, Johnson and Zhang, 2013], that take advantage of the finite sum structure of the problem to use cheaper iterations while preserving fast convergence. This paper aims at bridging the gap between stochastic

| ALGORITHM | SYNCHRONY | STOCHASTIC | TIME |
|---|---|---|---|
| POINT-SAGA [DEFAZIO, 2016] | N/A | ✓ | $nm + \sqrt{nm\kappa_s}$ |
| MSDA [SCAMAN ET AL., 2017] | GLOBAL | ✗ | $\sqrt{\kappa_b}\left(m + \frac{\tau}{\sqrt{\gamma}}\right)$ |
| ESDACD [HENDRIKX ET AL., 2019] | LOCAL | ✗ | $(m + \tau)\sqrt{\frac{\kappa_b}{\gamma}}$ |
| DSBA [SHEN ET AL., 2018] | GLOBAL | ✓ | $\left(m + \kappa_s + \gamma^{-1}\right)(1 + \tau)$ |
| ADFS (THIS PAPER) | LOCAL | ✓ | $m + \sqrt{m\kappa_s} + (1 + \tau)\sqrt{\frac{\kappa_s}{\gamma}}$ |

Table 1: Comparison of various state-of-the-art decentralized algorithms to reach accuracy $\varepsilon$ in regular graphs. Constant factors are omitted, as well as the $\log\left(\varepsilon^{-1}\right)$ factor in the TIME column. Reported runtime for Point-SAGA corresponds to running it on a single machine with $nm$ samples. To allow for direct comparison, we assume that computing a dual gradient of a function $f_i$ as required by MSDA and ESDACD takes time $m$, although it is generally more expensive than to compute $m$ separate proximal operators of single $f_{i,j}$ functions.

and decentralized algorithms when local functions are smooth and strongly convex. In the rest of this paper, following Scaman et al. [2017], we assume that nodes are linked by a communication network and can only exchange messages with their neighbours. We further assume that each communication takes time $\tau$ and that processing one sample, *i.e.*, computing the proximal operator for a *single* function $f_{i,j}$, takes time 1. The proximal operator of a function $f_{i,j}$ is defined by $\text{prox}_{\eta f_{i,j}}(x) = \arg\min_v \frac{1}{2\eta}\|v - x\|^2 + f_{i,j}(v)$. The condition number of the Laplacian matrix of the graph representing the communication network is denoted $\gamma$. This natural constant appears in the running time of many decentralized algorithms and is for instance of order $O(1)$ for the complete graph and $O(n^{-1})$ for the 2D grid. More generally, $\gamma^{-1/2}$ is typically of the same order as the diameter of the graph. Following notations from Xiao et al. [2019], we define the batch and stochastic condition numbers $\kappa_b$ and $\kappa_s$ (which are classical quantities in the analysis of finite sum optimization) such that for all $i$, $\kappa_b \geq M_i/\sigma_i$ where $M_i$ is the smoothness constant of the function $f_i$ and $\kappa_s \geq \kappa_i$, with $\kappa_i = 1 + \sum_{j=1}^m L_{i,j}/\sigma_i$ the stochastic condition number of node $i$. Although $\kappa_s$ is always bigger than $\kappa_b$, it is generally of the same order of magnitude, leading to the practical superiority of stochastic algorithms. The next paragraphs discuss the relevant state of the art for both distributed and stochastic methods, and Table 1 sums up the speeds of the main decentralized algorithms available to solve Problem (1). Although it is not a distributed algorithm, Point-SAGA [Defazio, 2016], an optimal single-machine algorithm, is also presented for comparison.

***Centralized* gradient methods.** A simple way to split work between nodes is to distribute gradient computations and to aggregate them on a parameter server. Provided the network is fast enough, this allows the system to learn from the datasets of $n$ workers in the same time one worker would need to learn from its own dataset. Yet, these approaches are very sensitive to stochastic delays, slow nodes, and communication bottlenecks. Asynchronous methods may be used [Recht et al., 2011, Leblond et al., 2017, Xiao et al., 2019] to address the first two issues, but computing gradients on older (or even inconsistent) versions of the parameter harms convergence [Chen et al., 2016]. Therefore, this paper focuses on decentralized algorithms, which are generally less sensitive to communication bottlenecks [Lian et al., 2017].

***Decentralized* gradient methods.** In their synchronous versions, decentralized algorithms alternate rounds of computations (in which all nodes compute gradients with respect to their local data) and communications, in which nodes exchange information with their direct neighbors [Duchi et al., 2012, Shi et al., 2015, Nedic et al., 2017, Tang et al., 2018, He et al., 2018]. Communication steps often consist in averaging gradients or parameters with neighbours, and can thus be abstracted as multiplication by a so-called gossip matrix. MSDA [Scaman et al., 2017] is a batch decentralized synchronous algorithm, and it is optimal with respect to the constants $\gamma$ and $\kappa_b$, among batch algorithms that can only perform these two operations. Instead of performing global synchronous updates, some approaches inspired from gossip algorithms [Boyd et al., 2006] use randomized pairwise communications [Nedic and Ozdaglar, 2009, Johansson et al., 2009, Colin et al., 2016]. This for example allows fast nodes to perform more updates in order to benefit from their increased computing power. These randomized algorithms do not suffer from the usual worst-case analyses of bounded-delay asynchronous algorithms, and can thus have fast rates because the step-size does not need to be reduced in the presence of delays. For example, ESDACD [Hendrikx et al., 2019] achieves the same optimal speed as MSDA when batch computations are faster than communications ($\tau > m$).

However, both use gradients of the Fenchel conjugates of the full local functions, which are generally much harder to get than regular gradients.

**Stochastic algorithms for finite sums.** All distributed methods presented earlier are *batch* methods that rely on computing *full gradient* steps of each function $f_i$. Stochastic methods perform updates based on randomly chosen functions $f_{i,j}$. In the smooth and strongly convex setting, they can be coupled with *variance reduction* [Schmidt et al., 2017, Shalev-Shwartz and Zhang, 2013, Johnson and Zhang, 2013, Defazio et al., 2014] and *acceleration*, to achieve the $m + \sqrt{m\kappa_s}$ optimal finite-sum rate, which greatly improves over the $m\sqrt{\kappa_b}$ batch optimum when the dataset is large. Examples of such methods include Accelerated-SDCA [Shalev-Shwartz and Zhang, 2014], APCG [Lin et al., 2015], Point-SAGA [Defazio, 2016] or Katyusha [Allen-Zhu, 2017]

**Decentralized stochastic methods.** In the smooth and strongly convex setting, DSA [Mokhtari and Ribeiro, 2016] and later DSBA [Shen et al., 2018] are two linearly converging stochastic decentralized algorithms. DSBA uses the proximal operator of individual functions $f_{i,j}$ to significantly improve over DSA in terms of rates. Yet, DSBA does not enjoy the $\sqrt{m\kappa_s}$ accelerated rates and needs an excellent network with very fast communications. Indeed, nodes need to communicate each time they process a single sample, resulting in many communication steps. CHOCO-SGD [Koloskova et al., 2019] is a simple decentralized stochastic algorithm with support for compressed communications. Yet, it is not linearly convergent and it requires to communicate between each gradient step as well. Therefore, to the best of our knowledge, there is no decentralized stochastic algorithm with accelerated linear convergence rate or low communication complexity without sparsity assumptions (*i.e.*, sparse features in linear supervised learning).

**ADFS.** The main contribution of this paper is a locally synchronous **A**ccelerated **D**ecentralized stochastic algorithm for **F**inite **S**ums, named ADFS. It is very similar to APCG for empirical risk minimization in the limit case $n = 1$ (single machine), for which it gets the same $m + \sqrt{m\kappa_s}$ rate. Besides, this rate stays unchanged when the number of machines grows, meaning that ADFS can process $n$ times more data in the same amount of time on a network of size $n$. This scaling lasts as long as $(1+\tau)\sqrt{\kappa_s}\gamma^{-\frac{1}{2}} < m + \sqrt{m\kappa_s}$. This means that ADFS is at least as fast as MSDA unless both the network is extremely fast (communications are faster than evaluating a single proximal operator) and the diameter of the graph is very large compared to the size of the local finite sums. Therefore, ADFS outperforms MSDA and DSBA in most standard machine learning settings, combining optimal network scaling with the efficient distribution of optimal sequential finite-sum algorithms. Note however that, similarly to DSBA and Point-SAGA, ADFS requires evaluating $\text{prox}_{f_{i,j}}$, which requires solving a local optimization problem. Yet, in the case of linear models such as logistic regression, it is only a constant factor slower than computing $\nabla f_{i,j}$, and it is especially much faster than computing the gradient of the conjugate of the full dual functions $\nabla f_i^*$ required by ESDACD and MSDA, which were not designed for finite sums on each node in the first place.

ADFS is based on three novel technical contributions: (i) a novel augmented graph approach which yields the dual formulation of Section 2, (ii) an extension of the APCG algorithm to arbitrary sampling that is applied to the dual problem in order to get the generic algorithm of Section 3, and (iii) the analysis of local synchrony, which is performed in Section 4. Finally, Section 5 presents a relevant choice of parameters leading to the rates shown in Table 1, and an experimental comparison is done in Section 6. A Python implementation of ADFS is also provided in supplementary material.

## 2   Model and Derivations

We now specify our approach to solve the problem in Equation (1). The first (classical) step consists in considering that all nodes have a local parameter, but that all local parameters should be equal because the goal is to have the global minimizer of the sum. Therefore, the problem writes:

$$\min_{\theta \in \mathbb{R}^{n \times d}} \sum_{i=1}^{n} f_i(\theta^{(i)}) \text{ such that } \theta^{(i)} = \theta^{(j)} \text{ if } j \in \mathcal{N}(i), \tag{2}$$

where $\mathcal{N}(i)$ represents the neighbors of node $i$ in the communication graph. Then, ESDACD and MSDA are obtained by applying accelerated (coordinate) gradient descent to an appropriate dual formulation of Problem (2). In the dual formulation, constraints become variables and so updating a dual coordinate consists in performing an update along an edge of the network. In this work, we consider a new virtual graph in order to get a stochastic algorithm for finite sums. The transformation

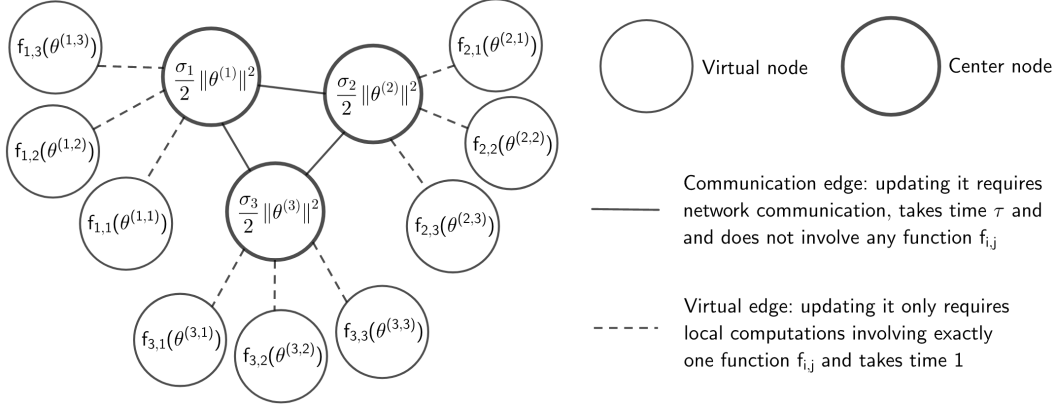

Figure 1: Illustration of the augmented graph for $n = 3$ and $m = 3$.

is sketched in Figure 1, and consists in replacing each node of the initial network by a star network. The centers of the stars are connected by the actual communication network, and the center of the star network replacing node $i$ has the local function $f_i^{\text{comm}} : x \mapsto \frac{\sigma_i}{2}\|x\|^2$. The center of node $i$ is then connected with $m$ nodes whose local functions are the functions $f_{i,j}$ for $j \in \{1, ..., m\}$. If we denote $E$ the number of edges of the initial graph, then the augmented graph has $n(1+m)$ nodes and $E + nm$ edges.

Then, we consider one parameter vector $\theta^{(i,j)}$ for each function $f_{i,j}$ and one vector $\theta^{(i)}$ for each function $f_i^{\text{comm}}$. Therefore, there is one parameter vector for each node in the augmented graph. We impose the standard constraint that the parameter of each node must be equal to the parameters of its neighbors, but neighbors are now taken in the augmented graph. This yields the following minimization problem:

$$\min_{\theta \in \mathbb{R}^{n(1+m) \times d}} \sum_{i=1}^{n} \left[ \sum_{j=1}^{m} f_{i,j}(\theta^{(i,j)}) + \frac{\sigma_i}{2}\|\theta^{(i)}\|^2 \right] \tag{3}$$

such that $\theta^{(i)} = \theta^{(j)}$ if $j \in \mathcal{N}(i)$, and $\theta^{(i,j)} = \theta^{(i)} \;\; \forall j \in \{1, .., m\}$.

In the rest of the paper, we use letters $k, \ell$ to refer to any nodes in the augmented graph, and letters $i, j$ to specifically refer to a communication node and one of its virtual nodes. More precisely, we denote $(k, \ell)$ the edge between the nodes $k$ and $\ell$ in the augmented graph. Note that $k$ and $\ell$ can be virtual or communication nodes. We denote $e^{(k)}$ the unit vector of $\mathbb{R}^{n(1+m)}$ corresponding to node $k$, and $e_{k\ell}$ the unit vector of $\mathbb{R}^{E+nm}$ corresponding to edge $(k, \ell)$. To clearly make the distinction between node variables and edge variables, for any matrix on the set of nodes of the augmented graph $x \in \mathbb{R}^{n(1+m) \times d}$ we write that $x^{(k)} = x^T e^{(k)}$ for $k \in \{1, ..., n(1+m)\}$ (superscript notation) and for any matrix on the set of edges of the augmented graph $\lambda \in \mathbb{R}^{(E+nm) \times d}$ we write that $\lambda_{k\ell} = \lambda^T e_{k\ell}$ (subscript notation) for any edge $(k, \ell)$. For node variables, we use the subscript notation with a $t$ to denote time, for instance in Algorithm 1. By a slight abuse of notations, we use indices $(i, j)$ instead of $(k, \ell)$ when specifically refering to virtual edges (or virtual nodes) and denote $\lambda_{ij}$ instead of $\lambda_{i,(i,j)}$ the virtual edge between node $i$ and node $(i, j)$ in the augmented graph. The constraints of Problem (3) can be rewritten $A^T \theta = 0$ in matrix form, where $A \in \mathbb{R}^{n(1+m) \times (nm+E)}$ is such that $Ae_{k\ell} = \mu_{k\ell}(e^{(k)} - e^{(\ell)})$ for some $\mu_{k\ell} > 0$. Then, the dual formulation of this problem writes:

$$\max_{\lambda \in \mathbb{R}^{(nm+E) \times d}} - \sum_{i=1}^{n} \left[ \sum_{j=1}^{m} f_{i,j}^* \left( (A\lambda)^{(i,j)} \right) + \frac{1}{2\sigma_i}\|(A\lambda)^{(i)}\|^2 \right], \tag{4}$$

where the parameter $\lambda$ is the Lagrange multiplier associated with the constraints of Problem (3)—more precisely, for an edge $(k, \ell)$, $\lambda_{k\ell} \in \mathbb{R}^d$ is the Lagrange multiplier associated with the constraint $\mu_{k\ell}(e^{(k)} - e^{(\ell)})^T \theta = 0$. At this point, the functions $f_{i,j}$ are only assumed to be convex (and not necessarily strongly convex) meaning that the functions $f_{i,j}^*$ are potentially non-smooth. This problem could be bypassed by transferring some of the quadratic penalty from the communication nodes to the virtual nodes before going to the dual formulation. Yet, this approach fails when $m$ is large because the smoothness parameter of $f_{i,j}^*$ would scale as $m/\sigma_i$ at best, whereas a smoothness of order $1/\sigma_i$

is required to match optimal finite-sum methods. A better option is to consider the $f_{i,j}^*$ terms as non-smooth and perform proximal updates on them. The rate of proximal gradient methods such as APCG [Lin et al., 2015] does not depend on the strong convexity parameter of the non-smooth functions $f_{i,j}^*$. Each $f_{i,j}^*$ is $(1/L_{i,j})$-strongly convex (because $f_{i,j}$ was $(L_{i,j})$-smooth), so we can rewrite the previous equation in order to transfer all the strong convexity to the communication node. Noting that $(A\lambda)^{(i,j)} = -\mu_{ij}\lambda_{ij}$ when node $(i,j)$ is a virtual node associated with node $i$, we rewrite the dual problem as:

$$\min_{\lambda \in \mathbb{R}^{(E+nm) \times d}} q_A(\lambda) + \sum_{i=1}^{n} \sum_{j=1}^{m} \tilde{f}_{i,j}^*(\lambda_{ij}), \tag{5}$$

with $\tilde{f}_{i,j}^* : x \mapsto f_{i,j}^*(-\mu_{ij}x) - \frac{\mu_{ij}^2}{2L_{i,j}}\|x\|^2$ and $q_A : x \mapsto \text{Trace}\big(\frac{1}{2}x^T A^T \Sigma^{-1} Ax\big)$, where $\Sigma$ is the diagonal matrix such that $e^{(i)^T}\Sigma e^{(i)} = \sigma_i$ if $i$ is a center node and $e^{(i,j)^T}\Sigma e^{(i,j)} = L_{i,j}$ if it is the virtual node $(i,j)$. Since dual variables are associated with edges, using coordinate descent algorithms on dual formulations from a well-chosen augmented graph of constraints allows us to handle both computations and communications in the same framework. Indeed, choosing a variable corresponding to an actual edge of the network results in a communication along this edge, whereas choosing a virtual edge results in a local computation step. Then, we balance the ratio between communications and computations by adjusting the probability of picking a given kind of edges.

## 3 The Algorithm: ADFS Iterations and Expected Error

In this section, we detail our new ADFS algorithm. In order to obtain it, we introduce a generalized version of the APCG algorithm [Lin et al., 2015], which we detail in Appendix A. More specifically, this generalized version allows for arbitrary sampling of coordinates, which is required to use different probabilities for communications and computations. It also includes corrections for functions that are strongly convex on a subspace only, which is the case of our augmented dual problem since the Laplacian of a graph is not full rank. Then we apply it to Problem (5) to get Algorithm 1. Due to lack of space, we only present the smooth version of ADFS here, but a non-smooth version is presented in Appendix B, along with the derivations required to obtain Algorithm 1 and Theorem 1. We denote $A^\dagger$ the pseudo inverse of $A$ and $W_{k\ell} \in \mathbb{R}^{n(1+m) \times n(1+m)}$ the matrix such that $W_{k\ell} = (e^{(k)} - e^{(\ell)})(e^{(k)} - e^{(\ell)})^T$ for any edge $(k,\ell)$. Note that variables $x_t$, $y_t$ and $v_t$ from Algorithm 1 are variables associated with the nodes of the augmented graph and are therefore matrices in $\mathbb{R}^{n(1+m) \times d}$ (one row for each node). They are obtained by multiplying the dual variables of the proximal coordinate gradient algorithm applied to the dual problem of Equation (5) by $A$ on the left. We denote $\sigma_A = \lambda_{\min}^+(A^T\Sigma^{-1}A)$ the smallest non-zero eigenvalue of the matrix $A^T\Sigma^{-1}A$.

---

**Algorithm 1** ADFS$(A, (\sigma_i), (L_{i,j}), (\mu_{k\ell}), (p_{k\ell}), \rho)$

---

1: $\sigma_A = \lambda_{\min}^+(A^T\Sigma^{-1}A)$, $\tilde{\eta}_{k\ell} = \frac{\rho\mu_{k\ell}^2}{\sigma_A p_{k\ell}}$, $R_{k\ell} = e_{k\ell}^T A^\dagger A e_{k\ell}$    // *Initialization*
2: $x_0 = y_0 = v_0 = z_0 = 0^{(n+nm) \times d}$
3: **for** $t = 0$ to $K - 1$ **do**    // *Run for K iterations*
4:     $y_t = \frac{1}{1+\rho}(x_t + \rho v_t)$
5:     Sample edge $(k,\ell)$ with probability $p_{k\ell}$    // *Edge sampled from the augmented graph*
6:     $z_{t+1} = v_{t+1} = (1 - \rho)v_t + \rho y_t - \tilde{\eta}_{k\ell} W_{k\ell} \Sigma^{-1} y_t$    // *Nodes k and $\ell$ communicate $y_t$*
7:     **if** $(k,\ell)$ is the virtual edge between node $i$ and virtual node $(i,j)$ **then**
8:         $v_{t+1}^{(i,j)} = \text{prox}_{\tilde{\eta}_{ij}\tilde{f}_{i,j}^*}\left(z_{t+1}^{(i,j)}\right)$    // *Virtual node update using $f_{i,j}$*
9:         $v_{t+1}^{(i)} = z_{t+1}^{(i)} + z_{t+1}^{(i,j)} - v_{t+1}^{(i,j)}$    // *Center node update*
10:    **end if**
11:    $x_{t+1} = y_t + \frac{\rho R_{k\ell}}{p_{k\ell}}(v_{t+1} - (1-\rho)v_t - \rho y_t)$
12: **end for**
13: **return** $\theta_K = \Sigma^{-1} v_K$    // *Return primal parameter*

---

**Theorem 1.** *We denote $\theta^\star$ the minimizer of the primal function $F : x \mapsto \sum_{i=1}^{n} f_i(x)$ and $\theta_A^\star$ a minimizer of the dual function $F_A^* = q_A + \psi$. Then $\theta_t$ as output by Algorithm 1 verifies:*

$$\mathbb{E}\left[\|\theta_t - \theta^\star\|^2\right] \leq C_0(1-\rho)^t, \quad \text{if} \quad \rho^2 \leq \min_{k\ell} \frac{\lambda_{\min}^+(A^T\Sigma^{-1}A)}{\Sigma_{kk}^{-1} + \Sigma_{\ell\ell}^{-1}} \frac{p_{k\ell}^2}{\mu_{k\ell}^2 R_{k\ell}}, \tag{6}$$

*with $C_0 = \lambda_{\max}(A^T\Sigma^{-2}A)\left[\|A^\dagger A\theta_A^\star\|^2 + 2\sigma_A^{-1}\left(F_A^*(0) - F_A^*(\theta_A^\star)\right)\right]$.*

We discuss several aspects related to the implementation of Algorithm 1 below, and provide its Python implementation in supplementary material.

**Convergence rate.** The parameter $\rho$ controls the convergence rate of ADFS. It is defined by the minimum of the individual rates for each edge, which explicitly depend on parameters related to the functions themselves $(1/(\Sigma_{kk}^{-1} + \Sigma_{\ell\ell}^{-1}))$, to the graph topology $(R_{k\ell} = e_{k\ell}^T A^\dagger A e_{k\ell})$, to a mix of both $(\lambda_{\min}^+(A^T\Sigma^{-1}A)/\mu_{k\ell}^2)$ and to the sampling probabilities of the edges $(p_{k\ell}^2)$. Note that these quantities are very different depending on whether edges are virtual or not. For example, the parameters $\mu_{k\ell}$ for communication edges are related to the communication matrix by the fact that the Laplacian of the communication network writes $L = \sum_{\text{communication } (k,\ell)} \mu_{k\ell}^2 W_{k\ell}$. In Section 5, we carefully choose the parameters $\mu_{k\ell}$ and $p_{k\ell}$ based on the graph and the local functions to get the best convergence speed. Note that once $\mu_{k\ell}$ and $p_{k\ell}$ are fixed, the choice of the other parameters (such as $R_{k\ell}$, $\rho$, $\eta$ and $\sigma_A$) is fixed as well (no extra tuning is required).

**Obtaining Line 6.** The form of the communication update (virtual or not) comes from the fact that the update in direction $(k, \ell)$ writes $A\nabla_{k\ell}q_A(y_t) = Ae_{k\ell}e_{k\ell}^T A\Sigma^{-1}y_t = \mu_{k\ell}^2 W_{k\ell}\Sigma^{-1}y_t$.

**Sparse updates.** Although the updates of Algorithm 1 involve all nodes of the network, it is actually possible to implement them efficiently so that only two nodes are actually involved in each update, as described below. Indeed, $W_{k\ell}$ is a very sparse matrix so $\left(W_{k\ell}\Sigma^{-1}y_t\right)^{(k)} = (\Sigma_k^{-1}y_t^{(k)} - \Sigma_\ell^{-1}y_t^{(\ell)}) = -\left(W_{k\ell}\Sigma^{-1}y_t\right)^{(\ell)}$ and $\left(W_{k\ell}\Sigma^{-1}y_t\right)^{(h)} = 0$ for $h \neq k, \ell$. Therefore, only the following situations can happen:

1. **Communication updates**: If $(k, \ell)$ is a communication edge, the update only requires nodes $k$ and $\ell$ to exchange parameters and perform a weighted difference between them. Note that the Laplacian of the communication graph is $\sum_{k\ell}$

2. **Local updates**: If $(k, \ell)$ is the virtual edge between node $i$ and its $j$-th virtual node, parameters exchange of line 4 is local, and the proximal term involves function $f_{i,j}$ only.

3. **Convex combinations**: If we choose $h \neq k, \ell$ then $v_{t+1}^{(h)}$ and $y_{t+1}^{(h)}$ are obtained by convex combinations of $y_t^{(h)}$ and $v_t^{(h)}$ so the update is cheap and local. Besides, nodes actually need the value of their parameters only when they perform updates of type 1 or 2. Therefore, they can simply store how many updates of this type they should have done and perform them all at once before each communication or local update.

**Primal proximal step.** Algorithm 1 uses proximal steps performed on $\tilde{f}_{i,j}^* : x \to f_{i,j}^*(-\mu_{i,j}x) - \frac{\mu_{i_j}^2}{2L_{i,j}}\|x\|^2$ instead of $f_{i,j}$. Yet, it is possible to use Moreau identity to express $\text{prox}_{\eta\tilde{f}_{i,j}^*}$ using only the proximal operator of $f_{i,j}$, which can easily be evaluated for many objective functions. The exact derivations are presented in Appendix B.3.

**Linear case.** For many standard machine learning problems, $f_{i,j}(\theta) = \ell(X_{i,j}^T\theta)$ with $X_{i,j} \in \mathbb{R}^d$. This implies that $f_{i,j}^*(\theta) = +\infty$ whenever $\theta \notin \text{Vec}(X_{i,j})$. Therefore, the proximal steps on the Fenchel conjugate only have support on $X_{i,j}$, meaning that they are one-dimensional problems that can be solved in constant time using for example the Newton method when no analytical solution is available. Warm starts (initializing on the previous solution) can also be used for solving the local problems even faster so that in the end, a one-dimensional proximal update is only a constant time slower than a gradient update. Note that this also allows to store parameters $v_t$ and $y_t$ as scalar coefficients for virtual nodes, thus greatly reducing the memory footprint of ADFS.

**Unbalanced local datasets.** We assume that all local datasets are of fixed size $m$ in order to ease reading. Yet, the impact of the value of $m$ on Algorithm 1 is indirect, and unbalanced datasets can be handled without any change. Yet, this may affect waiting time since nodes with large local datasets will generally be more busy than nodes with smaller ones.

# 4 Distributed Execution and Synchronization Time

Theorem 1 gives bounds on the expected error after a given number of iterations. To assess the actual speed of the algorithm, it is still required to know how long executing a given number of iterations takes. This is easy with synchronous algorithms such as MSDA or DSBA, in which all nodes iteratively perform local updates or communication rounds. In this case, executing $n_{\text{comp}}$ computing rounds and $n_{\text{comm}}$ communication rounds simply takes time $n_{\text{comp}} + \tau n_{\text{comm}}$. ADFS relies on randomized pairwise communications, so it is necessary to sample a *schedule*, *i.e.*, a random sequence of edges from the augmented graph, and evaluate how fast this schedule can be executed. Note that the execution time crucially depends on how many edges can be updated in parallel, which itself depends on the graph and on the random schedule sampled.

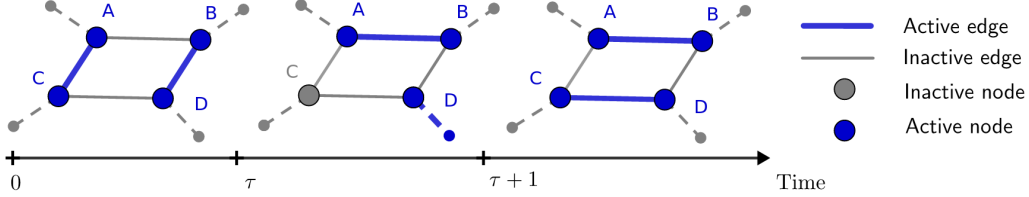

Figure 2: Illustration of parallel execution and local synchrony. Nodes from a toy graph execute the schedule $[(A, C), (B, D), (A, B), (D), (C, D)]$, where $(D)$ means that node $D$ performs a local update. Each node needs to execute its updates in the partial order defined by the schedule. In particular, node $C$ has to perform update $(A, C)$ and then update $(C, D)$, so it is idle between times $\tau$ and $\tau + 1$ because it needs to wait for node $D$ to finish its local update before the communication update $(C, D)$ can start. We assume $\tau > 1$ since the local update terminates before the communication update $(A, B)$. Contrary to synchronous algorithms, no global notion of rounds exist and some nodes (such as node $D$) perform more updates than others.

**Shared schedule.** Even though they only actively take part in a small fraction of the updates, all nodes need to execute the same schedule to correctly implement Algorithm 1. To generate this shared schedule, all nodes are given a seed and the sampling probabilities of all edges. This allows them to avoid deadlocks and to precisely know how many convex combinations to perform between $v_t$ and $y_t$.

**Execution time.** The problem of bounding the probability that a random schedule of fixed length exceeds a given execution time can be cast in the framework of fork-join queuing networks with blocking [Zeng et al., 2018]. In particular, queuing theory [Baccelli et al., 1992] tells us that the average time per iteration exists for any fixed probability distribution over a given augmented graph. Unfortunately, existing quantitative results are not precise enough for our purpose so we generalize the method introduced by Hendrikx et al. [2019] to get a finer bound. While their result is valid when the only possible operation is communicating with a neighbor, we extend it to the case in which nodes can also perform local computations. For the rest of this paper, we denote $p_{\text{comm}}$ the probability of performing a communication update and $p_{\text{comp}}$ the probability of performing a local update. They are such that $p_{\text{comp}} + p_{\text{comm}} = 1$. We also define $p_{\text{comm}}^{\max} = n \max_k \sum_{\ell \in \mathcal{N}(k)} p_{k\ell}/2$, where neighbors are in the communication network only. When all nodes have the same probability to participate in an update, $p_{\text{comm}}^{\max} = p_{\text{comm}}$. Then, the following theorem holds (see proof in Appendix C):

**Theorem 2.** *Let $T(t)$ be the time needed for the system to execute a schedule of size $t$, i.e., $t$ iterations of Algorithm 1. If all nodes perform local computations with probability $p_{\text{comp}}/n$ with $p_{\text{comp}} > p_{\text{comm}}^{\max}$ or if $\tau > 1$ then there exists $C < 24$ such that:*

$$\mathbb{P}\left(\frac{1}{t}T(t) \leq \frac{C}{n}\left(p_{\text{comp}} + 2\tau p_{\text{comm}}^{\max}\right)\right) \to 1 \text{ as } t \to \infty \qquad (7)$$

Note that the constant $C$ is a worst-case estimate and that it is much smaller for homogeneous communication probabilities. This novel result states that the number of iterations that Algorithm 1 can perform per unit of time increases linearly with the size of the network. This is possible because each iteration only involves two nodes so many iterations can be done in parallel. The assumption $p_{\text{comp}} > p_{\text{comm}}$ is responsible for the $1 + \tau$ factor instead of $\tau$ in Table 1, which prevents ADFS from benefiting from *network acceleration* when communications are cheap ($\tau < 1$). Note that this is an actual restriction of following a schedule, as detailed in Appendix C. Yet, network operations

generally suffer from communication protocols overhead whereas computing a single proximal update often either has a closed-form solution or is a simple one-dimensional problem in the linear case. Therefore, assuming $\tau > 1$ is not very restrictive in the finite-sum setting.

## 5 Performances and Parameters Choice in the Homogeneous Setting

We now prove the time to convergence of ADFS presented in Table 1, and detail the conditions under which it holds. Indeed, Section 3 presents ADFS in full generality but the different parameters have to be chosen carefully to reach optimal speed. In particular, we have to choose the coefficients $\mu$ to make sure that the graph augmentation trick does not cause the smallest positive eigenvalue of $A^T \Sigma^{-1} A$ to shrink too much. Similarly, $\rho$ is defined in Equation (6) by a minimum over all edges of a given quantity. This quantity heavily depends on whether the edge is an actual communication edge or a virtual edge. One can trade $p_{\text{comp}}$ for $p_{\text{comm}}$ so that the minimum is the same for both kind of edges, but Theorem 2 tells us that this is only possible as long as $p_{\text{comp}} > p_{\text{comm}}$.

**Parameters choice.** We define $L = A_{\text{comm}} A_{\text{comm}}^T \in \mathbb{R}^{n \times n}$ the Laplacian of the communication graph, with $A_{\text{comm}} \in \mathbb{R}^{n \times E}$ such that $A_{\text{comm}} e_{k\ell} = \mu_{k\ell}(e^{(k)} - e^{(\ell)})$ for all edge $(k, \ell) \in E^{\text{comm}}$, the set of communication edges. Then, we define $\tilde{\gamma} = \min_{(k,\ell) \in E^{\text{comm}}} \lambda^+_{\min}(L) n^2 / (\mu^2_{k\ell} R_{k\ell} E^2)$. As shown in Appendix D.2, $\tilde{\gamma} \approx \gamma$ for regular graphs such as the complete graph or the grid, justifying the use of $\gamma$ instead of $\tilde{\gamma}$ in Table 1. We assume for simplicity that $\sigma_i = \sigma$ and that $\kappa_i = 1 + \sigma_i^{-1} \sum_{j=1}^m L_{i,j} = \kappa_s$ for all nodes. For virtual edges, we choose $\mu^2_{ij} = \lambda^+_{\min}(L) L_{i,j} / (\sigma \kappa_i)$ and $p_{ij} = p_{\text{comp}}(1 + L_{i,j} \sigma_i^{-1})^{\frac{1}{2}} / (n S_{\text{comp}})$ with $S_{\text{comp}} = n^{-1} \sum_{i=1}^n \sum_{j=1}^m (1 + L_{i,j} \sigma_i^{-1})^{\frac{1}{2}}$. This corresponds to using a standard importance sampling scheme for selecting samples. For communications edges $(k, \ell) \in E^{\text{comm}}$, we choose uniform $p_{k\ell} = p_{\text{comm}} / E$ and $\mu^2_{k\ell} = 1/2$.

**Parameters tuning.** The previous paragraph specifies relevant choices of parameters $\mu_{k\ell}$ and $p_{k\ell}$. Therefore, ADFS can be run *without manual tuning*. Extra tuning (such as communication probabilities) could be performed to adapt to specific heterogeneous situations. Yet, this should be considered as an extra degree of freedom that other algorithms may not have access to rather than an extra parameter to tune. For example, the choice of uniform communication probabilities is automatically enforced by synchronous gossip-based algorithms such as MSDA or DSBA (all edges are activated at each step). Note that choosing different values of $\mu_{k\ell}$ for communication edges amounts to tuning the gossip matrix, which is generally considered as an input of the problem. Our specific choice of $\mu_{ij}$ for virtual edges allows to precisely bound the strong convexity of the augmented problem $\sigma_A$, as shown in Appendix D.1.

**Influence of the network topology.** The topology of the network only impacts the convergence rate through the constant $\tilde{\gamma}$, which is almost equal to the eigengap of the Laplacian of the graph for regular networks. This dependence is standard, as it can be seen in Table 1. The topology can also influence the synchronization time since the presence of hubs generally increases waiting time.

**Theorem 3.** *If we choose* $p_{\text{comm}} = \min\left(1/2, \left(1 + S_{\text{comp}}\sqrt{\tilde{\gamma}/\kappa_s}\right)^{-1}\right)$. *Then, running Algorithm 1 for* $K_\varepsilon = \rho^{-1} \log(\varepsilon^{-1})$ *iterations guarantees* $\mathbb{E}\left[\|\theta_{K_\varepsilon} - \theta^\star\|^2\right] \leq C_0 \varepsilon$, *and takes time* $T(K_\varepsilon)$, *with:*

$$T(K_\varepsilon) \leq \sqrt{2}C\left(m + \sqrt{m\kappa_s} + \sqrt{2}\left(1 + 4\tau\right)\sqrt{\frac{\kappa_s}{\tilde{\gamma}}}\right)\log\left(1/\varepsilon\right)$$

*with probability tending to* 1 *as* $\rho^{-1}\log(\varepsilon^{-1}) \to \infty$, *with $C_0$ and $C$ as in Theorems 1 and 2.*

Theorem 3 assumes that all communication probabilities and condition numbers are exactly equal in order to ease reading. A more detailed version with rates for more heterogeneous settings can be found in Appendix D. Note that while algorithms such as MSDA required to use polynomials of the initial gossip matrix to model several consecutive communication steps, we can more directly tune the amount of communication and computation steps simply by adjusting $p_{\text{comp}}$ and $p_{\text{comm}}$.

## 6 Experiments

In this section, we illustrate the theoretical results by showing how ADFS compares with MSDA [Scaman et al., 2017], ESDACD [Hendrikx et al., 2019], Point-SAGA [Defazio, 2016], and DSBA [Shen et al., 2018]. All algorithms (except for DSBA, for which we fine-tuned the step-size) were run

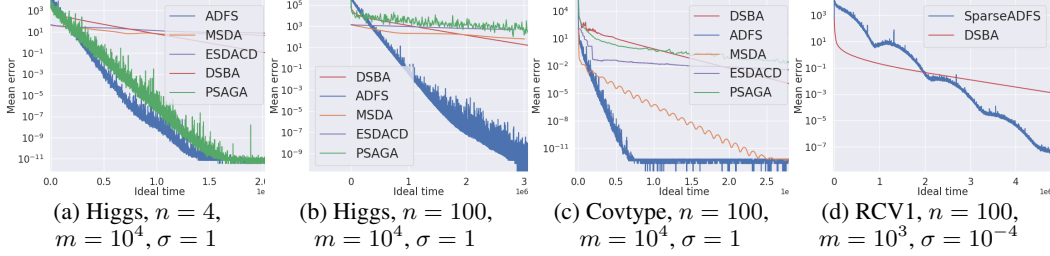

(a) Higgs, $n = 4$,
$m = 10^4$, $\sigma = 1$

(b) Higgs, $n = 100$,
$m = 10^4$, $\sigma = 1$

(c) Covtype, $n = 100$,
$m = 10^4$, $\sigma = 1$

(d) RCV1, $n = 100$,
$m = 10^3$, $\sigma = 10^{-4}$

Figure 3: Performances of various decentralized algorithms on the logistic regression task with $m = 10^4$ points per node, regularization parameter $\sigma = 1$ and communication delays $\tau = 5$ on 2D grid networks of different sizes.

with out-of-the-box hyperparameters given by theory on data extracted from the standard Higgs, Covtype and RCV1 datasets from LibSVM. The underlying graph is assumed to be a 2D grid network. Experiments were run in a distributed manner on an actual computing cluster. Yet, plots are shown for *idealized times* in order to abstract implementation details as well as ensure that reported timings were not impacted by the cluster status or implementation details. All the details of the experimental setup as well as a comparison with centralized algorithms can be found in Appendix E. An implementation of ADFS is also available in supplementary material.

Figure 3a shows that, as predicted by theory, ADFS and Point-SAGA have similar rates on small networks whereas all other algorithms are significantly slower. Figures 3b, 3c and 3d use a much larger grid to evaluate how these algorithms scale. In this setting, Point-SAGA is the slowest algorithm since it has 100 times less computing power available. MSDA performs quite well on the Covtype dataset thanks to its very good network scaling (dependent on $\kappa_b$ rather than $\kappa_s$). Yet, the $m\sqrt{\kappa_b}$ factor dominates on the Higgs dataset, making it significantly slower. DSBA has to communicate after each proximal step, thus having to wait for a time $\tau = 5$ at each step. ESDACD does not perform well either because $m > \tau$ and it has to perform as many batch computing steps as communication steps. ADFS does not suffer from any of these drawbacks and therefore outperforms other approaches by a large margin on these experiments. This illustrates the fact that ADFS combines the strengths of accelerated stochastic algorithms, such as Point-SAGA, and fast decentralized algorithms, such as MSDA. We see that DSBA initially outperforms ADFS on the RCV1 dataset. This may be due to statistical reasons, since there is more overlap of the local datasets of the different nodes in this experiment than in the others. Yet, we see that ADFS has a better rate in the steady state and quickly catches up. Besides, we still used a value $\tau = 5$ but a much higher value of $\tau$ would be more realistic in this high dimensional setting since local computations are sparse whereas communications are fully dimensional. We only compare DSBA and ADFS in this setting since the high-dimensionality of the dataset made the computation of dual gradients expensive, and Point-SAGA is much slower when using 100 nodes since it is a single-machine algorithm, as shown on the Higgs and Covtype datasets.

# 7    Conclusion

In this paper, we provided a novel accelerated stochastic algorithm for decentralized optimization. To the best of our knowledge, it is the first decentralized algorithm that successfully leverages the finite-sum structure of the objective functions to match the rates of the best known sequential algorithms while having the network scaling of optimal batch algorithms. The analysis in this paper could be extended to better handle heterogeneous settings, both in terms of hardware (computing times, delays) and local functions (different regularities). Finally, finding a locally synchronous algorithm that can take advantage of arbitrarily low communication delays (beyond the $\tau > 1$ limit) to scale to large graphs is still an open problem.

# Acknowledgement

We acknowledge support from the European Research Council (grant SEQUOIA 724063).

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
