[Supplementary Material]

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

Section A is a self-contained section with the statement and proofs of the extended APCG algorithm. Then, Section B presents the derivations required to obtain ADFS from the extended APCG algorithm. Section C is dedicated to the study of waiting time in the locally synchronous model, and the analysis of the speed of ADFS for a specific choice of parameters is then given in Section D. Finally, Section E details the experimental setting and gives additional experiments involving centralized algorithms.

## A Generalized APCG

In this section, we study the generic problem of accelerated proximal coordinate descent. We give an algorithm that works with *arbitrary sampling* of the coordinates, thus yielding a stronger result than state-of-the-art approaches [Lin et al., 2015, Fercoq and Richtárik, 2015]. This is a key contribution that allows to obtain fast rates when sampling probabilities are heterogeneous and determined by the problem. It is especially useful in our case to pick different probabilities for computing and for communicating. We also extend the result to the case in which the function is strongly convex only on a subspace. Since this section of the Appendix is intended to detail the extended APCG general algorithm for a generic problem, it is mostly self-contained, and notations are in particular different from the rest of the paper. More specifically, we study the following generic problem:

$$\min_{x \in \mathbb{R}^d} \ f_A(x) + \sum_{i=1}^{d} \psi_i(x^{(i)}), \tag{8}$$

where all the functions $\psi_i$ are convex and $f_A$ is such that there exists a matrix $A$ such that $f_A$ is $(\sigma_A)$-strongly convex on $\mathrm{Ker}(A)^{\perp}$, the orthogonal of the kernel of $A$. Since, $A^{\dagger}A$ is the projector on $\mathrm{Ker}(A)^{\perp}$, (recall that $A^{\dagger}$ is the pseudo-inverse of $A$), the strong convexity on this subspace can be written as the fact that for all $x, y \in \mathbb{R}^d$:

$$f_A(x) - f_A(y) \geq \nabla f_A(y)^T A^{\dagger} A(x - y) + \tfrac{\sigma_A}{2}(x - y)^T A^{\dagger} A(x - y). \tag{9}$$

Note that this implies that $f_A$ is constant on $\mathrm{Ker}(A)$, so in particular there exists a function $f$ such that for any $x \in \mathbb{R}^d$, $f_A(x) = f(Ax)$. In this case, $\sigma_A$ is such that $x^T A^T \nabla^2 f(y) Ax \geq \sigma_A \|x\|^2$ for any $x \in \mathrm{Ker}(A)^{\perp}$ and $y \in \mathbb{R}^d$. Besides, $f_A$ is assumed to be $(M_i)$-smooth in direction $i$ meaning that its gradient in the direction $i$ (noted $\nabla_i f_A$) is $(M_i)$-Lipschitz. This is the general setting of the problem of Equation (5), that can be recovered by taking $f_A = q_A$ and $\psi_i = \tilde{f}_i^*$. Proximal coordinate gradient algorithms are known to work well for these problems, which is why we would like to use APCG [Lin et al., 2014]. Yet, we would like to pick different probabilities for computing and communication edges, whereas APCG only handles uniform coordinates sampling. Furthermore, the first term is strongly-convex only on the orthogonal of the kernel of the matrix $A$, so APCG cannot be applied straightforwardly. Therefore, we introduce an extended version of APCG, presented in Algorithm 2, and we explicit its rate in Theorem 4. This extended APCG can then directly be applied to solve the problem of Equation (5).

### A.1 Algorithm and results

In this appendix, since there is no need to distinguish between primal and dual variables variables as in the main text, we denote $e_i \in \mathbb{R}^d$ the unit vector corresponding to coordinate $i$, and $x^{(i)} = e_i^T x$ for any $x \in \mathbb{R}^d$. Let $R_i = e_i^T A^{\dagger} A e_i$ and $p_i$ be the probability that coordinate $i$ is picked to be updated. Constant $S$ is such that $S^2 \geq \frac{M_i R_i}{p_i^2}$ for all $i$. Then, following the approaches of Nesterov and Stich [2017] and Lin et al. [2015], we fix $A_0, B_0 \in \mathbb{R}$ and recursively define sequences $\alpha_t, \beta_t, a_t, A_t$ and $B_t$ such that:

$$a_{t+1}^2 S^2 = A_{t+1} B_{t+1}, \qquad B_{t+1} = B_t + \sigma_A a_{t+1}, \qquad A_{t+1} = A_t + a_{t+1},$$
$$\alpha_t = \frac{a_{t+1}}{A_{t+1}}, \qquad \beta_t = \frac{\sigma_A a_{t+1}}{B_{t+1}}.$$

Finally, we introduce the sequences $(y_t)$, $(v_t)$ and $(x_t)$, that are all initialized at 0, and $(w_t)$ such that for all $t$, $w_t = (1 - \beta_t)v_t + \beta_t y_t$. We define $\eta_{i,t} = \frac{a_{t+1}}{B_{t+1}p_i}$ and the proximal operator:

$$\mathrm{prox}_{\eta_{i,t}\psi_i} : x \mapsto \arg\min_v \frac{1}{2\eta_{i,t}}\|v - x\|^2 + \psi_i(v).$$

We denote $\nabla_i f_A = e_i e_i^T \nabla f_A$ the coordinate gradient of $f_A$ along direction $i$. For generalized

---

**Algorithm 2** Generalized APCG($A_0, B_0, S, \sigma_A$)

---

$y_0 = 0, v_0 = 0, t = 0$
**while** $t < T$ **do**
$\quad y_t = \frac{(1-\alpha_t)x_t + \alpha_t(1-\beta_t)v_t}{1 - \alpha_t\beta_t}$
$\quad$ Sample $i$ with probability $p_i$
$\quad v_{t+1} = z_{t+1} = (1-\beta_t)v_t + \beta_t y_t - \eta_{i,t}\nabla_i f_A(y_t)$
$\quad v_{t+1}^{(i)} = \text{prox}_{\eta_i \psi_i}\left(z_{t+1}^{(i)}\right)$
$\quad x_{t+1} = y_t + \frac{\alpha_t R_i}{p_i}(v_{t+1} - (1-\beta_t)v_t - \beta_t y_t)$
**end while**

---

APCG to work well, the proximal operator needs to be taken in the subspace defined by the projector $A^\dagger A$, and so the non-smooth $\psi_i$ terms have to be separable after composition with $A^\dagger A$. Since $A^\dagger A$ is a projector, this constraint is equivalent to stating that either $R_i = 1$ (projection does not affect the coordinate $i$), or $\psi_i = 0$ (no proximal update to make).

**Assumption 1.** *The functions $f_A$ and $\psi$ are such that equation Equation (9) holds for some $\sigma_A \geq 0$ and for all $i \in \mathbb{R}^d$, $f_A$ is $(M_i)$-smooth in direction $i$ and $\psi$ and $A$ are such that either $R_i = 1$ or $\psi_i = 0$.*

This natural assumption allows us to formulate the proximal update in standard squared norm since the proximal operator is only used for coordinates $i$ for which $A^\dagger A e_i = e_i$. Then, we formulate Algorithm 2 and analyze its rate in Theorem 4.

**Theorem 4.** *Let $F : x \mapsto f_A(x) + \sum_{i=1}^d \psi_i\left(x^{(i)}\right)$ such that Assumption 1 holds. If $S$ is such that $S^2 \geq \frac{M_i R_i}{p_i^2}$ and $1 - \beta_t - \frac{\alpha_t R_i}{p_i} \geq 0$, the sequences $v_t$ and $x_t$ generated by APCG verify:*

$$B_t \mathbb{E}\left[\|v_t - \theta^\star\|_{A^\dagger A}^2\right] + 2A_t\left[\mathbb{E}\left[F(x_t)\right] - F(\theta^\star)\right] \leq C_0,$$

*where $C_0 = B_0\|v_0 - \theta^\star\|^2 + 2A_0\left[F(x_0) - F(\theta^\star)\right]$ and $\theta^\star$ is a minimizer of $F$. The rate of APCG depends on $S$ through the sequences $\alpha_t$ and $\beta_t$.*

Our extended APCG algorithm is also closely related with an arbitrary sampling version of AP-PROX [Fercoq and Richtárik, 2015]. Yet, APPROX has an explicit formulation with a more flexible block selection rule than choosing only one coordinate at a time. Similarly to Lee and Sidford [2013], it also uses iterations that can be more efficient, especially in the linear case. These extensions can also be applied to APCG under the same assumptions, but this is beyond the scope of this paper. Theorem 4 is a general method that in particular requires to set values for $A_0, B_0, \alpha_0$ and $\beta_0$. The two following corollaries give choices of parameters depending on whether $\sigma_A > 0$ or $\sigma_A = 0$, along with the rate of APCG in these cases.

**Corollary 1** (Strongly Convex case). *Let $F$ be such that it verifies the assumptions of Theorem 4. If $\sigma_A > 0$, we can choose for all $t \in \mathbb{N}$ $\alpha_t = \beta_t = \rho$ and $A_t = \sigma_A^{-1}B_t = (1-\rho)^{-t}$ with $\rho = \sqrt{\sigma_A}S^{-1}$. In this case, the condition $1 - \beta_t - \frac{\alpha_t R_i}{p_i} \geq 0$ can be weakened to $1 - \frac{\alpha_t R_i}{p_i} \geq 0$ and it is automatically satisfied by our choice of $S$, $\alpha_t$ and $\beta_t$. In this case, APCG converges linearly with rate $\rho$, as shown by the following result:*

$$\sigma_A \mathbb{E}\left[\|v_t - \theta^\star\|_{A^\dagger A}^2\right] + 2\left[\mathbb{E}\left[F(x_t)\right] - F(\theta^\star)\right] \leq C_0(1-\rho)^t$$

**Corollary 2** (Convex case). *Let $F$ be such that it verifies the assumptions of Theorem 4. If $\sigma_A = 0$, we can choose $\beta_t = 0$ and $\alpha_0 = p_{\min} = \min_i p_i$. In this case, the condition $1 - \beta_t - \frac{\alpha_t R_i}{p_i} \geq 0$ is always satisfied for our choice of $S$ and the error verifies:*

$$\mathbb{E}\left[F(x_t)\right] - F(\theta^\star) \leq \frac{2}{t^2}\left[S^2 r_t^2 + \frac{2}{p_{\min}^2}\left[F(x_0) - F(\theta^\star)\right]\right],$$

*with $r_t^2 = \|v_0 - \theta^\star\|_{A^\dagger A}^2 - \mathbb{E}[\|v_t - \theta^\star\|_{A^\dagger A}^2]$.*

In the convex case, we only have control over the objective function $F$ and not over the parameters. This in particular means that it is only possible to have guarantees on the dual objective in the case of non-smooth ADFS.

## A.2 Proof of Theorem 4

Before starting the proof, we define $w_t = (1 - \beta_t)v_t + \beta_t y_t$, and:

$$V_i^t(v) = \frac{B_{t+1}p_i}{2a_{t+1}}\|v - w_t^{(i)} + \eta_i e_i^T \nabla f(y_t)\|^2 + \psi_i(v).$$

Then, we give the following lemma, that we prove later:

**Lemma 1.** *If either $1 - \beta_t - \frac{\alpha_t}{p_i} \geq 0$ or $\alpha_t = \beta_t$ and $1 - \frac{\alpha_t}{p_i} \geq 0$ for any $i$ such that $\psi_i \neq 0$, then for any $t$ and $i$ such that $\psi_i \neq 0$, we can write $x_t^{(i)} = \sum_{l=0}^t \delta_t^{(i)}(l)v_l^{(i)}$ such that $\sum_{l=0}^t \delta_t^{(i)}(l) = 1$ and for any $l$, $\delta_t^{(i)}(l) \geq 0$. We define $\hat{\psi}_t^{(i)} = \sum_{l=0}^t \delta_t^{(i)}(l)\psi_i(v_l^{(i)})$ and $\hat{\psi}_t = \sum_{i=1}^d \hat{\psi}_t^{(i)}$. Then, if $R_i = 1$ whenever $\psi_i \neq 0$, $\psi(x_t) \leq \hat{\psi}_t$ and:*

$$\mathbb{E}_{i_t}\left[\hat{\psi}_{t+1}\right] \leq \alpha_t \psi(\tilde{v}_{t+1}) + (1 - \alpha_t)\hat{\psi}_t. \tag{10}$$

*where $\tilde{v}_{t+1}^{(i)} = \arg\min_v V_i^t(v)$ for all $i$. In particular, $v_{t+1}^{(i_t)} = \tilde{v}_{t+1}^{(i_t)}$ and $v_{t+1}^{(j)} = w_t^{(j)}$ for $j \neq i_t$.*

Note that Lemma 1 is a generalization to arbitrary sampling probabilities of the beginning of the proof in Lin et al. [2015]. We can now prove the main theorem.

*Proof of Theorem 4.* This proof follows the same general structure as Nesterov and Stich [2017]. In particular, it follows from expanding the $\|v_{t+1} - \theta^\star\|^2$ term. In the original proof, $v_{t+1} = w_t - g$ where $g$ is a gradient term so the expansion is rather straightforward. In our case, $v_{t+1}$ is defined by a proximal mapping so a bit more work is required. Yet, similar terms will appear, plus the function values of the non-smooth term that we control with Lemma 1. We start by showing the following equality:

$$\frac{B_{t+1}p_i}{2a_{t+1}}[\|v_{t+1} - \theta^\star\|_{A^\dagger A}^2 + \|v_{t+1} - w_t\|_{A^\dagger A}^2 - \|\theta^\star - w_t\|_{A^\dagger A}^2]$$

$$\leq \langle \nabla_i f_A(y_t), \theta^\star - v_{t+1}\rangle_{A^\dagger A} + \psi_i\left(\theta^{\star(i)}\right) - \psi_i\left(v_{t+1}^{(i)}\right). \tag{11}$$

When $\psi_i = 0$, it follows from using $v_{t+1} = w_t - \frac{a_{t+1}}{B_{t+1}p_i}\nabla_i f_A(y_t)$ and basic algebra (expanding the squared terms).

When $\psi_i \neq 0$, $A^\dagger A e_i = e_i$ because $e_i^T A^\dagger A e_i = 1$ and $A^\dagger A$ is a projector. Therefore, we obtain

$$\|v_{t+1} - \theta^\star\|_{A^\dagger A}^2 - \|w_t - \theta^\star\|_{A^\dagger A}^2 = \|v_{t+1}^{(i)} - \theta^{\star(i)}\|^2 - \|w_t^{(i)} - \theta^{\star(i)}\|^2, \tag{12}$$

because $v_{t+1}$ is equal to $w_t$ for coordinates other than $i$. We now use the strong convexity of $V_i^t$ at points $v_{t+1}^{(i)}$ (its minimizer, by definition) and $\theta^{\star(i)}$ ($i$-th coordinate of a minimizer of $F$) to write that $V_i^t(v_{t+1}^{(i)}) + \frac{B_{t+1}p_i}{2a_{t+1}}\|v_{t+1}^{(i)} - \theta^{\star(i)}\|^2 \leq V_i^t(\theta^{\star(i)})$. This is a key step from the proof of Lin et al. [2015]. Then, expanding the $V_i^t$ terms yields:

$$\frac{B_{t+1}p_i}{2a_{t+1}}\left[\|v_{t+1}^{(i)} - \theta^{\star(i)}\|^2 + \|v_{t+1}^{(i)} - w_t^{(i)} + \frac{a_{t+1}}{B_{t+1}p_i}\nabla_i f_A(y_t)\|^2 - \|\theta^\star - w_t + \frac{a_{t+1}}{B_{t+1}p_i}\nabla_i f_A(y_t)\|^2\right]$$

$$\leq \psi_i\left(\theta^{\star(i)}\right) - \psi_i\left(v_{t+1}^{(i)}\right).$$

We can now retrieve Equation (11) by pulling gradient terms out of the squares and using Equation (12). We now evaluate each term of Equation (11). First of all, we use the form of $x_{t+1}$ and the fact that $w_t - v_{t+1} = e_i^T(w_t - v_{t+1})$ (only one coordinate is updated) to show:

$$\mathbb{E}\left[\frac{a_{t+1}}{p_i}\langle \nabla_i f_A(y_t), \theta^\star - v_{t+1}\rangle_{A^\dagger A}\right] = a_{t+1}\mathbb{E}\left[\langle\frac{1}{p_i}\nabla_i f_A(y_t), \theta^\star - w_t\rangle_{A^\dagger A}\right]$$

$$+ A_{t+1}\mathbb{E}\left[\langle\nabla_i f_A(y_t), \frac{\alpha_t}{p_i}(w_t - v_{t+1})\rangle_{A^\dagger A}\right]$$

$$= a_{t+1}\langle\nabla f_A(y_t), \theta^\star - w_t\rangle_{A^\dagger A} + A_{t+1}\mathbb{E}\left[\langle\nabla_i f_A(y_t), y_t - x_{t+1}\rangle\right],$$

where we used that $R_i = e_i^T A^\dagger A e_i$ and $y_t - x_{t+1} = \frac{\alpha_t R_i}{p_i}(w_t - v_{t+1})$.

The rest of this proof closely follows the analysis from Hendrikx et al. [2019], which is an adaptation of Nesterov and Stich [2017] to strong convexity only on a subspace. The main difference is that it is also necessary to control the function values of $\psi$, which is done using Lemma 1. For the first term, we use the strong convexity of $f$ as well as the fact that $w_t = y_t - \frac{1-\alpha_t}{\alpha_t}(x_t - y_t)$ to obtain:

$$a_{t+1}\nabla f_A(y_t)^T A^\dagger A(\theta^\star - w_t) = a_{t+1}\nabla f_A(y_t)^T A^\dagger A\left(\theta^\star - y_t + \frac{1-\alpha_t}{\alpha_t}(x_t - y_t)\right)$$

$$\leq a_{t+1}\left(f_A(\theta^\star) - f_A(y_t) - \frac{1}{2}\sigma_A\|y_t - \theta^\star\|_{A^\dagger A}^2 + \frac{1-\alpha_t}{\alpha_t}(f_A(x_t) - f_A(y_t))\right)$$

$$\leq a_{t+1}f_A(\theta^\star) - A_{t+1}f_A(y_t) + A_t f_A(x_t) - \frac{1}{2}a_{t+1}\sigma_A\|y_t - \theta^\star\|_{A^\dagger A}^2.$$

For the second term, we use the fact that $x_{t+1} - y_t$ has support on $e_i$ only (just like $v_{t+1} - w_t$) and the directional smoothness of $f_A$ to obtain:

$$A_{t+1}\langle \nabla_i f_A(y_t), y_t - x_{t+1}\rangle \leq A_{t+1}\left[f_A(y_t) - f_A(x_{t+1}) + \frac{M_i}{2}\|x_{t+1} - y_t\|^2\right]$$

$$\leq A_{t+1}(f_A(y_t) - f_A(x_{t+1})) + \frac{B_{t+1}}{2}\frac{M_i R_i}{p_i^2}\frac{a_{t+1}^2}{A_{t+1}B_{t+1}}R_i\|e_i^T(v_{t+1} - w_t)\|^2$$

$$\leq A_{t+1}(f_A(y_t) - f_A(x_{t+1})) + \frac{B_{t+1}}{2}\|v_{t+1} - w_t\|_{A^\dagger A}^2.$$

Noting $\Delta f_A(x_t) = \mathbb{E}[f(x_t)] - f_A(\theta^\star)$ and remarking that $a_{t+1} = A_{t+1} - A_t$, we obtain, using that $\alpha_t = \frac{a_{t+1}}{A_{t+1}}$:

$$\mathbb{E}\left[\frac{a_{t+1}}{p_i}\langle \nabla_i f_A(y_t), \theta^\star - v_{t+1}\rangle_{A^\dagger A}\right] \leq A_t\Delta f_A(x_t) - A_{t+1}\Delta f_A(x_{t+1}) + \frac{B_{t+1}}{2}\mathbb{E}\left[\|w_t - v_{t+1}\|_{A^\dagger A}^2\right]$$

$$- \frac{a_{t+1}\sigma_A}{2}\|y_t - \theta^\star\|_{A^\dagger A}^2.$$

Using Lemma 1, we derive in the same way:

$$\mathbb{E}\left[\frac{a_{t+1}}{p_i}\left[\psi_i\left(\theta^{\star(i)}\right) - \psi_i\left(v_{t+1}^{(i)}\right)\right]\right] = a_{t+1}\psi(\theta^\star) - A_{t+1}\alpha_t\psi(\tilde{v}_{t+1})$$

$$\leq A_t\left(\hat{\psi}_t - \psi(\theta^\star)\right) - A_{t+1}\left(\hat{\psi}_{t+1} - \psi(\theta^\star)\right).$$

Now, we can multiply Equation (11) by $\frac{a_{t+1}}{p_i}$ and take the expectation over $i$. The $\|v_{t+1} - w_t\|_{A^\dagger A}^2$ terms cancel and we obtain:

$$\frac{B_{t+1}}{2}\mathbb{E}\left[\|v_{t+1} - \theta^\star\|_{A^\dagger A}^2\right] + A_{t+1}\Delta\hat{F}_A(x_{t+1}) \leq$$

$$A_t\Delta\hat{F}_A(x_t) + \frac{B_{t+1}}{2}\|w_t - \theta^\star\|_{A^\dagger A}^2 - \frac{a_{t+1}\sigma_A}{2}\|y_t - \theta^\star\|_{A^\dagger A}^2,$$

where $\Delta\hat{F}_A(x_t) = \Delta f_A(x_t) + \mathbb{E}\left[\hat{\psi}_t\right] - \psi(\theta^\star)$. Convexity of the squared norm yields $\|w_t - \theta^\star\|_{A^\dagger A}^2 \leq (1 - \beta_t)\|v_t - \theta^\star\|_{A^\dagger A}^2 + \beta_t\|y_t - \theta^\star\|_{A^\dagger A}^2$. Now remarking that $B_{t+1}(1 - \beta_t) = B_t$ and $a_{t+1}\sigma_A = B_{t+1}\beta_t$, and summing the inequalities until $t = 0$, we obtain:

$$B_t\mathbb{E}\left[\|v_t - \theta^\star\|_{A^\dagger A}^2\right] + 2A_t\Delta\hat{F}_A(x_t) \leq 2A_0\Delta F_A(x_0) + B_0\|v_0 - \theta^\star\|_{A^\dagger A}^2.$$

We finish the proof by using the fact that $\psi(x_t) \leq \hat{\psi}_t$ and $\psi(x_0) = \hat{\psi}_0$ since $x_0 = v_0$. □

Now that we have proven Theorem 4, we can proceed to the proof of Lemma 1.

*Proof of Lemma 1.* This lemma is a generalization of the lemma from APCG with arbitrary probabilities (instead of uniform ones). It still uses the fact that $x_t$ can be written as a convex combination of $(v_l)_{l \leq t}$, but it requires to use a different convex combination for each coordinate of $x_t$, thus crucially exploiting the separability of the proximal term. If coordinate $i$ is such that $\psi_i = 0$, then $\hat{\psi}_{t+1}^{(i)} \leq \alpha_t\psi_i(\tilde{v}_{t+1}^{(i)}) + (1 - \alpha_t)\hat{\psi}_t^{(i)}$ is automatically satisfied for any $\delta_t^{(i)}$. For coordinates $i$ such that

$\psi_i \neq 0$ (and so $R_i = 1$), we start by expressing $x_{t+1}$ in terms of $x_t$, $v_{t+1}$ and $v_t$. More precisely, we write that for any $t > 0$:

$$x_{t+1}^{(i)} = y_t^{(i)} + \frac{\alpha_t}{p_i}(v_{t+1}^{(i)} - w_t^{(i)}).$$

Indeed, either coordinate $i$ is updated at time $t$ or $v_{t+1}^{(i)} = w_t^{(i)}$ so the previous equation always holds. We can then develop the $w_t$ and $y_t$ terms to obtain $x_{t+1}^{(i)}$ only in function of $x_t^{(i)}$, $v_t^{(i)}$ and $v_{t+1}^{(i)}$:

$$
\begin{aligned}
x_{t+1}^{(i)} &= \frac{\alpha_t}{p_i}v_{t+1}^{(i)} + \left(1 - \frac{\alpha_t\beta_t}{p_i}\right)y_t^{(i)} - \frac{\alpha_t(1 - \beta_t)}{p_i}v_t^{(i)} \\
&= \frac{\alpha_t}{p_i}v_{t+1}^{(i)} + \left(1 - \frac{\alpha_t\beta_t}{p_i}\right)\frac{(1 - \alpha_t)x_t^{(i)} + \alpha_t(1 - \beta_t)v_t^{(i)}}{1 - \alpha_t\beta_t} - \frac{\alpha_t(1 - \beta_t)}{p_i}v_t^{(i)} \\
&= \frac{\alpha_t}{p_i}v_{t+1}^{(i)} + \alpha_t(1 - \beta_t)\left[\frac{1 - \frac{\alpha_t\beta_t}{p_i}}{1 - \alpha_t\beta_t} - \frac{1}{p_i}\right]v_t^{(i)} + \left(1 - \frac{\alpha_t\beta_t}{p_i}\right)\frac{(1 - \alpha_t)}{1 - \alpha_t\beta_t}x_t^{(i)} \\
&= \frac{\alpha_t}{p_i}v_{t+1}^{(i)} + \frac{\alpha_t(1 - \beta_t)}{1 - \alpha_t\beta_t}\left(1 - \frac{1}{p_i}\right)v_t^{(i)} + \left(1 - \frac{\alpha_t\beta_t}{p_i}\right)\frac{(1 - \alpha_t)}{1 - \alpha_t\beta_t}x_t^{(i)}.
\end{aligned}
$$

At this point, all coefficients sum to 1. Indeed, they all sum to 1 at the first line and we have expressed $w_t^{(i)}$ and then $y_t^{(i)}$ as convex combinations of other terms, thus keeping the value of the sum unchanged. Yet, $p_i < 1$ so the coefficient on the second term is negative. Fortunately, it is possible to show that the $v_t^{(i)}$ term in the decomposition of $x_t^{(i)}$ is large enough so that the $v_t^{(i)}$ term in the decomposition of $x_{t+1}^{(i)}$ is positive. More precisely, we now show by recursion that for $t \geq 0$:

$$x_{t+1}^{(i)} = \frac{\alpha_t}{p_i}v_{t+1}^{(i)} + \sum_{l=0}^{t}\delta_{t+1}^{(i)}(l)v_l^{(i)}, \tag{13}$$

with $\delta_{t+1}^{(i)}(l) \geq 0$ for $l \leq t$. For $t = 0$, $x_0 = v_0$ and $x_1^{(i)} = \frac{\alpha_0}{p_i}v_1^{(i)} + \left(1 - \frac{\alpha_0}{p_i}\right)v_0^{(i)}$. We now assume that Equation (13) holds for a given $t > 0$, and expand $\delta_{t+1}^{(i)}(t)$ to show that it is positive. Using that $\delta_t^{(i)}(t) = \frac{\alpha_t}{p_i}$, we write:

$$
\begin{aligned}
\delta_{t+1}^{(i)}(t) &= \frac{\alpha_t(1 - \beta_t)}{1 - \alpha_t\beta_t}\left(1 - \frac{1}{p_i}\right) + \frac{\alpha_t}{p_i}\left(1 - \frac{\alpha_t\beta_t}{p_i}\right)\frac{(1 - \alpha_t)}{1 - \alpha_t\beta_t} \\
&= \frac{\alpha_t}{1 - \alpha_t\beta_t}\left[(1 - \beta_t)\left(1 - \frac{1}{p_i}\right) + \frac{(1 - \alpha_t)}{p_i}\left(1 - \frac{\alpha_t\beta_t}{p_i}\right)\right] \\
&= \frac{\alpha_t}{1 - \alpha_t\beta_t}\left[1 - \beta_t - \frac{1}{p_i} + \frac{\beta_t}{p_i} + \frac{1}{p_i} - \frac{\alpha_t}{p_i} - (1 - \alpha_t)\frac{\alpha_t\beta_t}{p_i^2}\right] \\
&= \frac{\alpha_t}{1 - \alpha_t\beta_t}\left[\left(1 - \beta_t - \frac{\alpha_t}{p_i}\right) + \frac{\beta_t}{p_i}\left(1 - (1 - \alpha_t)\frac{\alpha_t}{p_i}\right)\right].
\end{aligned}
$$

We conclude that $\delta_{t+1}^{(i)}(t) \geq 0$ since $1 - \beta_t - \frac{\alpha_t}{p_i} \geq 0$. Note that this condition can be weakened to $1 - \frac{\alpha_t^2}{p_i^2} \geq 0$ when $\beta_t = \alpha_t$ or when $\beta_t = 0$. We also deduce from the form of $x_{t+1}^{(i)}$ that for $l < t$, the only coefficients on $v_l^{(i)}$ in the development of $x_{t+1}^{(i)}$ come from the $x_t^{(i)}$ term and so:

$$\delta_{t+1}^{(i)}(l) = \left(1 - \frac{\alpha_t\beta_t}{p_i}\right)\frac{(1 - \alpha_t)}{1 - \alpha_t\beta_t}\delta_t^{(i)}(l), \tag{14}$$

so these coefficients are positive as well. Since they also sum to 1, it implies that $x_t^{(i)}$ is a convex combination of the $v_l^{(i)}$ for $l \leq t$, and we use the convexity of $\psi_i$ to write:

$$\psi_i(x_t^{(i)}) = \psi_i\left(\sum_{l=0}^{t}\delta_t^{(i)}(l)v_l^{(i)}\right) \leq \sum_{l=0}^{t}\delta_t^{(i)}(l)\psi_i(v_l^{(i)}) = \hat{\psi}_t^{(i)}.$$

Now, we can properly express $\hat{\psi}_{t+1}^{(i)}$ using the decomposition of $x_{t+1}^{(i)}$ in terms of $\delta_{t+1}^{(i)}$:

$$\mathbb{E}\left[\hat{\psi}_{t+1}^{(i)}\right] = \mathbb{E}\left[\frac{\alpha_t}{p_i}\psi_i(v_{t+1}^{(i)})\right] + \frac{\alpha_t(1-\beta_t)}{1-\alpha_t\beta_t}\left(1-\frac{1}{p_i}\right)\psi_i(v_t^{(i)}) + \left(1-\frac{\alpha_t\beta_t}{p_i}\right)\frac{1-\alpha_t}{1-\alpha_t\beta_t}\sum_{l=0}^{t}\delta_t^{(i)}(l)\psi_i(v_l^{(i)})$$

$$= \alpha_t\psi_i(\tilde{v}_{t+1}^{(i)}) + (1-p_i)\frac{\alpha_t}{p_i}\psi_i(w_t^{(i)}) + \frac{\alpha_t(1-\beta_t)}{1-\alpha_t\beta_t}\left(1-\frac{1}{p_i}\right)\psi_i(v_t^{(i)}) + \left(1-\frac{\alpha_t\beta_t}{p_i}\right)\frac{1-\alpha_t}{1-\alpha_t\beta_t}\hat{\psi}_t^{(i)}$$

At this point, we use the convexity of $\psi_i$ to develop $\psi_i(w_t^{(i)})$ and then $\psi_i(y_t^{(i)})$ in the following way:

$$\psi_i(w_t^{(i)}) \leq (1-\beta_t)\psi_i(v_t^{(i)}) + \beta_t\psi_i(y_t^{(i)})$$

$$\leq (1-\beta_t)\psi_i(v_t^{(i)}) + \frac{\beta_t}{1-\alpha_t\beta_t}\left[(1-\alpha_t)\psi_i(x_t^{(i)}) + \alpha_t(1-\beta_t)\psi_i(v_t^{(i)})\right]$$

$$= \frac{1-\beta_t}{1-\alpha_t\beta_t}\psi_i(v_t^{(i)}) + \frac{\beta_t(1-\alpha_t)}{1-\alpha_t\beta_t}\psi_i(x_t^{(i)}).$$

If we plug these expressions into the development of $\mathbb{E}\left[\hat{\psi}_{t+1}^{(i)}\right]$, the $\psi_i(v_t^{(i)})$ terms cancel and we obtain:

$$\mathbb{E}\left[\hat{\psi}_{t+1}^{(i)}\right] \leq \alpha_t\psi_i(\tilde{v}_{t+1}^{(i)}) + \alpha_t\left(\frac{1}{p_i}-1\right)\frac{\beta_t(1-\alpha_t)}{1-\alpha_t\beta_t}\psi_i(x_t^{(i)}) + \left(1-\frac{\alpha_t\beta_t}{p_i}\right)\frac{1-\alpha_t}{1-\alpha_t\beta_t}\hat{\psi}_t^{(i)}$$

We now use the fact that $\psi_i(x_t^{(i)}) \leq \hat{\psi}_t^{(i)}$ (by convexity of $\psi_i$) to get:

$$\mathbb{E}\left[\hat{\psi}_{t+1}^{(i)}\right] \leq \alpha_t\psi_i(\tilde{v}_{t+1}^{(i)}) + \frac{1-\alpha_t}{1-\alpha_t\beta_t}\left[\alpha_t\beta_t\left(\frac{1}{p_i}-1\right) + \left(1-\frac{\alpha_t\beta_t}{p_i}\right)\right]\hat{\psi}_t^{(i)}$$

$$\leq \alpha_t\psi_i(\tilde{v}_{t+1}^{(i)}) + (1-\alpha_t)\hat{\psi}_t^{(i)}$$

This holds for any coordinate $i$ and so $\mathbb{E}\left[\hat{\psi}_{t+1}\right] \leq \alpha_t\psi(\tilde{v}_{t+1} + (1-\alpha_t)\hat{\psi}_t$ for all $t \geq 0$, which finishes the proof of the lemma. □

### A.3 Proof of the corollaries

Now that that we have proven the main result, we show how specific choices of parameters lead to fast algorithms.

*Proof of Corollary 1.* If $\sigma_A > 0$, then the parameters can be chosen as $\alpha_t = \beta_t = \rho = \frac{\sqrt{\sigma_A}}{S}$, with $A_t = (1-\rho)^{-t}$ and $B_t = \sigma_A A_t$. These expressions can then be plugged into the recursion to verify that they do satisfy it. This choice is classic and slightly suboptimal for small values of $t$ compared with the choice made by Nesterov and Stich [2017].

Yet, it remains to prove that $\alpha_t R_i \leq p_i$ to verify the assumptions of Theorem 4. This assumption was directly verified in the case of APCG thanks to the uniform probabilities. We show that this also holds in the arbitrary-sampling formulation with strong convexity on a subspace, and this result validates our choice of parameters. In particular, we write:

$$\alpha_t R_i = \rho R_i \leq \frac{\sqrt{\sigma_A}}{S}R_i \leq \sqrt{\frac{\sigma_A R_i}{M_i}}p_i.$$

Then, we take $x^\star$ such that $\nabla f_A(x^\star) = 0$ and use the smoothness and $(A^\dagger A)$-strong convexity of $f_A$ to write that for any coordinate $i$ and $h > 0$:

$$\frac{M_i}{2}\|he_i\|^2 \geq f(x^\star + he_i) - f(x^\star) \geq \frac{\sigma_A}{2}\|he_i\|_{A^\dagger A}.$$

In particular, this means that $M_i \geq \sigma_A R_i$, which means that $\alpha_t R_i \leq p_i$ for all $i$. □

*Proof of Corollary 2.* We first prove that $\alpha_t$ can actually be obtained by a simple recursion. This comes from the fact that the recursions in Lin et al. [2015] and Nesterov and Stich [2017] are actually the same. If $\sigma_A = 0$ then we have to choose $\beta_t = 0$ for all $t$. Then, we can choose $B_t = B_0$ for any $B_0 > 0$. This allows to write $(A_{t+1} - A_t)^2 S^2 = A_t B_0$ for all $t$, which is a second degree polynomial in the variable $A_{t+1}$. We choose the positive root in order to have $a_{t+1} \geq 0$, which yields:

$$A_{t+1} = A_t + \frac{B_0}{2S^2}\left(1 + \sqrt{1 + 4S^2 B_0^{-1} A_t}\right).$$

Coefficients $(a_t)$ can be computed using

$$a_{t+1} = A_{t+1} - A_t = \frac{B_0}{2S^2}\left(1 + \sqrt{1 + 4S^2 B_0^{-1} A_t}\right),$$

and so we use the fact that $a_{t+1} S^2 = A_{t+1} B_{t+1}$, which can be rewritten as $\alpha_t = \frac{B_0}{a_{t+1} S^2}$. to obtain the sequence $(\alpha_t)$ as:

$$\alpha_t = \frac{2}{1 + \sqrt{1 + 4S^2 B_0^{-1} A_t}}.$$

In particular,

$$A_t = \left[\left(\frac{2}{\alpha_t} - 1\right)^2 - 1\right]\frac{B_0}{4S^2}.$$

This expression for $A_t$ and $A_{t+1}$ can be substituted in the relation $A_{t+1} = A_t + \frac{B_0}{a_{t+1} S^2}$, which yields after some simplifications:

$$\alpha_{t+1}^{-2} - \alpha_{t+1}^{-1} - \alpha_t^{-2} = 0,$$

which is a second degree polynomial in the variable $\alpha_{t+1}^{-1}$. This can be solved, leading to

$$\alpha_{t+1} = \frac{2}{1 + \sqrt{1 + 4\alpha_t^{-2}}}.$$

Multiplying and dividing by $1 - \sqrt{1 + 4\alpha_t^{-2}}$ leads to:

$$\alpha_{t+1} = \frac{\sqrt{\alpha_t^4 + 4\alpha_t^2} - \alpha_t^2}{2},$$

which is the exact same recursion as in Lin et al. [2015] and Fercoq and Richtárik [2015]. In particular, only the value of $\alpha_0$ matters and only the sequence $\alpha_t$ actually needs to be computed, since the only coefficients needed are the $\alpha_t$ and $\frac{a_{t+1}}{B_{t+1}} = \frac{1}{\alpha_t S^2}$.

We would like to choose the smallest possible $\alpha_0$, se we take $\alpha_0 = p_{\min}$ where $p_{\min} = \min_i p_i$ where the minimum is over all coordinates such that $\psi_i \neq 0$. This is enough to respect the condition $\alpha_t \leq p_{\min}$ since $(\alpha_t)$ is a decreasing sequence. This leads to

$$A_0 = \left[\left(\frac{2}{p_{\min}} - 1\right)^2 - 1\right]\frac{B_0}{4S^2} \leq \frac{B_0}{p_{\min}^2 S^2}.$$

Since $A_0 \geq 0$, a direct recursion yields $A_t \geq \frac{B_0 t^2}{4S^2}$. We call $r_t^2 = \|v_0 - \theta_A^\star\|_{A^\dagger A}^2 - \mathbb{E}[\|v_t - \theta_A^\star\|_{A^\dagger A}^2]$, and $\Delta F_t = \mathbb{E}[F(x_t)] - F(\theta_A^\star)$, then:

$$\Delta F_t \leq \frac{1}{2A_t}\left(B_0 r_t^2 + 2A_0 F_0\right) = \frac{B_0}{2A_t}\left(r_t^2 + \frac{2}{S^2 p_{\min}^2}\Delta F_0\right) \leq \frac{2S^2}{t^2}\left(r_t^2 + \frac{2}{S^2 p_{\min}^2}\Delta F_0\right),$$

which finishes the proof of the rate. $\qquad\square$

# B Algorithm Derivation

## B.1 Projection of virtual edges

Theorem A requires that for any coordinate $i$, either the proximal part $\psi_i = 0$ or the coordinate is such that $e_i^T A^\dagger A e_i = 1$, which is equivalent to having $A^\dagger A e_i = e_i$. In our case, $\psi_{k\ell} = 0$ when $(k,\ell)$ is a communication edge. Lemma 2 is a small result that shows that the projection condition is satisfied by virtual edges.

**Lemma 2.** *If $(k,\ell)$ is a virtual edge then $R_{k\ell} = 1$.*

*Proof.* Let $x \in \mathbb{R}^{E+nm}$ such that $Ax = 0$. From the definition of $A$, either $x = 0$ or the support of $x$ is a cycle of the graph. Indeed, for any edge $(k,\ell)$, $Ae_{k\ell}$ has non-zero weights only on nodes $k$ and $\ell$. Virtual nodes have degree one, so virtual edges are parts of no cycles and therefore $x^T e_{k,\ell} = 0$ for all virtual edges $(k,\ell)$. Operator $A^\dagger A$ is the projection operator on the orthogonal the kernel of $A$, so it is the identity on virtual edges. $\qquad\square$

## B.2 From edge variables to node variables

Taking the dual formulation implies that variables are associated with edges rather than nodes. Although it could be possible to work with edge variables, it is generally inefficient. Indeed, the algorithm needs variable $Ay_t$ instead of variable $y_t$ for the gradient computation so standard methods work directly with $Ay_t$ [Scaman et al., 2017, Hendrikx et al., 2019].

In this section, we call $\tilde{v}_t$, $\tilde{y}_t$ and $\tilde{z}_t$ the dual variable sequences in $\mathbb{R}^{E+nm}$ obtained by applying Algorithm 2 on the dual problem of Equation 5. The new update equations can be retrieved by multiplying each line of Algorithm 2 by $A$ on the left, so that for example $v_t = A\tilde{v}_t$. Yet, there is still a $\tilde{z}_{t+1}$ term because of the presence of the proximal update. More specifically, we write for the virtual edge between node $i$ and its $j$-th virtual node:

$$\tilde{v}_{t+1}^T e_{ij} = \text{prox}_{\eta_{ij}\psi_{i,j}}\left(\tilde{z}_{t+1}^T e_{ij}\right). \tag{15}$$

Fortunately, this update only modifies $\tilde{v}_{t+1}$ when $\psi_{i,j} \neq 0$. This means that $z_{t+1}$ is only modified for local computation edges. Since local computation nodes only have one neighbour, the form of $A$ ensures that for any $\tilde{z} \in \mathbb{R}^{n(1+m)}$ and virtual edge $(k,\ell)$ corresponding to node $i$ and its $j$-th virtual node, $(A\tilde{z})^{(i,j)} = -\mu_{k\ell}\tilde{z}_{k\ell}$. In particular, if node $k$ is the center node $i$ and node $\ell$ is the virtual node $(i,j)$, the proximal update can be rewritten:

$$
\begin{aligned}
(A\tilde{v}_{t+1})^{(i,j)} &= -\mu_{ij}\text{prox}_{\eta_{ij}\psi_{i,j}}\left(-\frac{1}{\mu_{ij}}(A\tilde{z}_{t+1})^{(i,j)}\right) \\
&= -\mu_{ij}\arg\min_v \frac{1}{2\eta_{ij}}\|v - \left(-\frac{1}{\mu_{ij}}(A\tilde{z}_{t+1})^{(i,j)}\right)\|^2 + \psi_{i,j}(v) \\
&= -\mu_{ij}\arg\min_v \frac{1}{2\eta_{ij}\mu_{ij}^2}\|-\mu_{ij}v - (A\tilde{z}_{t+1})^{(i,j)}\|^2 + f_{i,j}^*(-\mu_{ij}v) - \frac{\mu_{ij}^2}{2L_{i,j}}\|v\|^2 \\
&= \arg\min_{\tilde{v}} \frac{1}{2\eta_{ij}\mu_{ij}^2}\|\tilde{v} - (A\tilde{z}_{t+1})^{(i,j)}\|^2 + f_{i,j}^*(\tilde{v}) - \frac{1}{2L_{i,j}}\|\tilde{v}\|^2 \\
&= \text{prox}_{\eta_{ij}\mu_{ij}^2 \tilde{f}_{i,j}^*}\left((A\tilde{z}_{t+1})^{(i,j)}\right),
\end{aligned}
$$

where $\tilde{f}_{i,j}^* : x \to f_{i,j}^*(x) - \frac{1}{2L_{i,j}}\|x\|^2$. For the center node, the update can be written:

$$
\begin{aligned}
(A\tilde{v}_{t+1})^{(i)} &= (A\tilde{z}_{t+1})^{(i)} - \mu_{ij}e_{ij}^T\tilde{z}_{t+1} + \mu_{ij}\text{prox}_{\eta_{ij}\psi_{i,j}}\left(-\frac{1}{\mu_{ij}}(A\tilde{z}_{t+1})^{(i,j)}\right) \\
&= (A\tilde{z}_{t+1})^{(i)} + (A\tilde{z}_{t+1})^{(i,j)} - \text{prox}_{\eta_{ij}\mu_{k\ell}^2 \tilde{f}_{i,j}^*}\left((A\tilde{z}_{t+1})^{(i,j)}\right).
\end{aligned}
$$

## B.3 Primal proximal updates

Moreau identity [Parikh and Boyd, 2014] provides a way to retrieve the proximal operator of $f^*$ using the proximal operator of $f$, but this does not directly apply to $\tilde{f}_{i,j}^*$, making its proximal update hard to compute when no analytical formula is available to compute $\tilde{f}_{i,j}^*$. Fortunately, the proximal

operator of $\tilde{f}_{i,j}^*$ can be retrieved from the proximal operator of $f_{i,j}^*$. More specifically, if we denote $\tilde{\eta}_{ij} = \eta_{ij}\mu_{ij}^2$ (it is clear in this section that they refer to the edge between node $i$ and its virtual node $j$), then we can also express the update only in terms of $f_{i,j}^*$:

$$\text{prox}_{\tilde{\eta}_{ij}\tilde{f}_{i,j}^*}\left((A\tilde{z}_{t+1})^{(i,j)}\right) = \arg\min_v \frac{1}{2\tilde{\eta}_{ij}}\|v - (A\tilde{z}_{t+1})^{(i,j)}\|^2 + f_{i,j}^*(v) - \frac{1}{2L_{i,j}}\|v\|^2$$

$$= \arg\min_v \frac{1}{2}\left(\tilde{\eta}_{ij}^{-1} - L_{i,j}^{-1}\right)\|v\|^2 - \tilde{\eta}_{ij}^{-1}v^T(A\tilde{z}_{t+1})^{(i,j)} + f_{i,j}^*(v)$$

$$= \arg\min_v \frac{1}{2\left(\tilde{\eta}_{ij}^{-1} - L_{i,j}^{-1}\right)^{-1}}\|v - \left(1 - \tilde{\eta}_{ij}L_{i,j}^{-1}\right)^{-1}(A\tilde{z}_{t+1})^{(i,j)}\|^2 + f_{i,j}^*(v)$$

$$= \text{prox}_{\left(\tilde{\eta}_{ij}^{-1} - L_{i,j}^{-1}\right)^{-1}f_{i,j}^*}\left(\left(1 - \tilde{\eta}_{ij}L_{i,j}^{-1}\right)^{-1}(A\tilde{z}_{t+1})^{(i,j)}\right).$$

Then, we use the identity:

$$\text{prox}_{(\eta f)^*}(x) = \eta\text{prox}_{\eta^{-1}f^*}\left(\eta^{-1}x\right), \tag{16}$$

and the Moreau identity to write that:

$$\text{prox}_{\eta f^*}(x) = x - \eta\text{prox}_{\eta^{-1}f}\left(\eta^{-1}x\right). \tag{17}$$

This allows us to retrieve the proximal operator on $\tilde{f}_{i,j}^*$ using only the proximal operator on $f_{i,j}$:

$$\left(1 - \tilde{\eta}_{ij}L_{i,j}^{-1}\right)\text{prox}_{\tilde{\eta}_{ij}\tilde{f}_{i,j}^*}\left((A\tilde{z}_{t+1})^{(i,j)}\right) = (A\tilde{z}_{t+1})^{(i,j)} - \tilde{\eta}_{ij}\text{prox}_{\left(\tilde{\eta}_{ij}^{-1} - L_{i,j}^{-1}\right)f}\left(\tilde{\eta}_{ij}^{-1}(A\tilde{z}_{t+1})^{(i,j)}\right). \tag{18}$$

Note that the previous calculations are valid as long as $\tilde{\eta}_{ij}L_{i,j}^{-1} \le 1$ for all virtual edges. A way to bound this is to replace by the values of $\mu_{ij}^2$ and $\sigma_A$ to get:

$$\rho \le \frac{\kappa_i}{2\kappa}p_{ij}.$$

The constraint $\rho < \min_{ij} p_{ij}$ was already enforced by APCG, so this simply gives another constraint that is generally verified unless nodes have very different local objectives (which should not happen if $m$ is big enough).

## B.4  Smooth case

If the functions $f_{i,j}$ are smooth then the functions $f_{i,j}^*$ are strongly convex and so function $q_A$ is strongly convex. ADFS can then be obtained by applying Algorithm 2 to Problem (5). The value of $S$ is obtained by remarking that $q_A$ is $\mu_{ij}^2\left(\Sigma_i^{-1} + \Sigma_j^{-1}\right)$ smooth in the direction $(i,j)$ and $\lambda_{\min}^+\left(A^T\Sigma^{-1}A\right)$ strongly convex on the orthogonal of the kernel of $A$. Lemma 2 guarantees that either $R_i = 1$ (virtual edges) or $\psi_i = 0$ (communication edges), so we can apply Corollary 1 to get:

$$B_t\mathbb{E}\left[\|\tilde{v}_t - \theta_A^\star\|_{A^\dagger A}^2\right] + 2A_t\left[\mathbb{E}\left[F_A^*(A\tilde{x}_t)\right] - F_A^*(\theta_A^\star)\right] \le C_0,$$

where $\tilde{v}_t$ and $\tilde{x}_t$ are the dual variables and $C_0$ is the same as in Theorem 1. ADFS works with variables $v_t = A\tilde{v}_t$ and $x_t = A\tilde{x}_t$ instead. Then, we use the fact that for any $x$, $F_A^*(x) = F_A^*(A^\dagger Ax)$ to write that $\mathbb{E}\left[F_A^*(\tilde{x}_t)\right] = \mathbb{E}\left[F_A^*(A^\dagger x_t)\right]$. Following Lin et al. [2015], and noting $q : x \mapsto \frac{1}{2}x^T\Sigma^{-1}x$ the primal optimal point $\theta^\star$ can be retrieved as $\theta^\star = \nabla q(A\theta_A^\star) = \Sigma^{-1}A\theta_A^\star$, where $\theta_A^\star$ is the optimal dual parameter. Finally,

$$\lambda_{\max}(A^T\Sigma^{-2}A)^{-1}\|\theta_t - \theta^\star\|^2 \le \lambda_{\max}(A^T\Sigma^{-2}A)^{-1}\|\Sigma^{-1}A(\tilde{v}_t - \theta_A^\star)\|^2 \le \|\tilde{v}_t - \theta_A^\star\|_{A^\dagger A}^2,$$

which finishes the proof of Theorem 1. Note that APCG also gives a guarantee in terms of dual function values at points $x_t$ but we drop it in order to have a simpler statement.

## B.5  Non-smooth setting

Extended APCG can be applied to the problem of Equation (5) even if function $q_A$ is not strongly convex on the orthogonal of the Kernel of $A$. This is for example the case when the functions $f_{i,j}$ are not smooth so that $\Sigma^{-1}$ has diagonal entries equal to 0 and therefore $\text{Ker}(A^T\Sigma^{-1}A) \not\subset \text{Ker}(A)$ so $\sigma_A = 0$. In this case, the choice of coefficients from Corollary 2 leads to Algorithm 3, a formulation of ADFS that provides error guarantees when primal functions $f_{i,j}$ are not smooth. More formally, if we define $F^* : x \to \sum_{i=1}^n\left[\sum_{j=1}^m f_{i,j}^*\left(x^{(i,j)}\right) + \frac{1}{2\sigma_i}\|x^{(i)}\|^2\right]$, then:

**Theorem 5.** *If the functions $f_{i,j}$ are non-smooth then NS-ADFS guarantees:*

$$\mathbb{E}\left[F^*(x_t)\right] - F^*(\theta^\star) \leq \frac{2}{t^2}\left[\frac{S^2}{\lambda_{\min}^+(A^T A)}r_t^2 + \frac{2}{p_{\min}^2}\left[F^*(x_0) - F^*(\theta^\star)\right]\right],$$

*with $r_t^2 = \|v_0 - \theta^\star\|^2 - \|v_t - \theta^\star\|^2$.*

The guarantees provided by Theorem 5 are weaker than in the smooth setting. In particular, we lose linear convergence and get the classical accelerated sublinear $O(1/t^2)$ rate. We also lose the bound on the primal parameters— recovering primal guarantees is beyond the scope of this work.

---
**Algorithm 3** NS-ADFS
---

$x_0 = 0$, $v_0 = 0$, $t = 0$, $\alpha_0 = \min_{\text{virtual edges }(i,j)} p_{ij}$, $\eta_{k\ell} = \frac{1}{\alpha_t S^2}$

**while** $t < T$ **do**

    $y_t = (1 - \alpha_t)x_t + \alpha_t v_t$

    Sample $(k, \ell)$ with probability $p_{k\ell}$

    $v_{t+1} = z_{t+1} = v_t - \eta_t \frac{\mu_{k\ell}^2}{p_{k\ell}} W_{k\ell} \Sigma^{-1} y_t$

    **if** $(k, \ell)$ is a computation edge between node $i$ and virtual node $(i, j)$ **then**

        $v_{t+1}^{(i,j)} = \text{prox}_{\eta_t \mu_{ij}^2 p_{ij}^{-1} f_{i,j}^*}\left(z_{t+1}^{(i,j)}\right)$

        $v_{t+1}^{(i)} = z_{t+1}^{(i)} + z_{t+1}^{(i,j)} - v_{t+1}^{(i,j)}$

    **end if**

    $x_{t+1} = y_t + \frac{\alpha_t R_{k\ell}}{p_{k\ell}}(v_{t+1} - v_t)$

    $\alpha_{t+1} = \frac{\sqrt{\alpha_t^4 + 4\alpha_t^2} - \alpha_t^2}{2}$

**end while**

**return** $\theta_t = \Sigma^{-1} v_t$

---

Note that the extra $\lambda_{\min}^+(A^T A)$ term comes from the fact that Theorem 5 is formulated with primal parameter sequences $x_t = A\tilde{x}_t$. Also note that $\alpha_t = O\left(t^{-1}\right)$, and $\frac{a_{t+1}}{B_{t+1}} = O\left(t\right)$. The leading constant governing the convergence rate is $\frac{\lambda_{\min}^+\left(A^T A\right)}{S^2}$, which is very related to the constant for the smooth case, simply that the $\Sigma^{-1}$ factor is removed. Therefore, we can obtain in the same way that if we choose $\mu_{ij}^2 = \frac{\lambda_{\min}^+(L)}{1+m}$ when $(i, j)$ is a computation edge then we get:

$$\lambda_{\min}^+(A^T A) \geq \frac{\lambda_{\min}^+(L)}{2(m+1)}.$$

Optimizing parameter $\rho$ in order to minimize time yields $\rho_{\text{comp}} = \rho_{\text{comm}}$ again, now leading in the homogeneous case to choosing:

$$p_{\text{comm}}^* = \left(1 + \sqrt{\frac{\tilde{\gamma}m^2}{2(1+m)}}\right)^{-1}.$$

## C  Average Time per Iteration

### C.1  More communications implies more waiting

A fundamental assumption for Theorem 2 is to assume that $p_{\text{comm}} < p_{\text{comp}}$. In particular, it prevents $p_{\text{comm}}$ from being too high since $p_{\text{comm}} + p_{\text{comp}} = 1$. Although this assumption seems quite restrictive in the first place, it is very intuitive to want to avoid $p_{\text{comm}}$ from being too high, especially in the limit of $p_{\text{comm}} \to 1$ and $\tau$ arbitrarily small. Consider that one node (say node $0$) starts a local update at some point. Communications are very fast compared to computations so it is very likely that the neighbors of node $0$ will only perform communication updates, and they will do so until they have to perform one with node $0$. At this point, they will have to wait until node $0$ finishes its local computation, which can take a long time. Now that the neighbors of node $0$ are also blocked waiting for the computation to finish, their neighbors will start establishing a dependence on them rather quickly. If the probability of computing is small enough and if the computing time is large enough, all nodes will sooner or later need to wait for node $0$ to finish its local update before they

can continue with the execution of their part of the schedule. In the end, only node 0 will actually be performing computations while all the others will be waiting.

This phenomenon is not restricted to the limit case presented above and the synchronization cost blows up as soon as $p_{\text{comm}} > p_{\text{comp}}$ and $\tau < 1$. In the proof below, the goal is to bound the total expected weight $\sum_{i=1}^{n} \mathbb{E}\left[X^t(i, w)\right]$ for $w$ higher than a given threshold. Local computing operations will move mass from small values of $w$ to higher values of $w$. On the other hand, communication operations will introduce synchronization between two nodes, thus increasing the total available mass $\sum_{w \geq 0} \sum_{i=1}^{n} \mathbb{E}\left[X^t(i, w)\right]$ (and not just moving it to higher values of $w$) because it will duplicate the mass for $X^t(i, w)$ to $X^t(j, w)$ if nodes $i$ and $j$ communicate. This is the technical reason why $p_{\text{comm}} < p_{\text{comp}}$ is needed for this proof.

### C.2 Detailed average time per iteration proof

The goal of this section is to prove Theorem 2. The proof is an extension of the proof of Theorem 2 from Hendrikx et al. [2019]. Similarly, we denote $t$ the number of iterations that the algorithm performs and $\tau_c^{ij}$ the random variable denoting the time taken by a communication on edge $(i, j)$. Similarly, $\tau_l^i$ denotes the time taken by a local computation at node $i$. Then, we introduce the random variable $X^t(i, w)$ such that if edge $(i, j)$ is activated at time $t + 1$ (with probability $p_{ij}$), then for all $w \in \mathbb{N}^*$:

$$X^{t+1}(i, w) = X^t(i, w - \tau_c^{ij}(t)) + X^t(j, w - \tau_c^{ij}(t)),$$

where $\tau_c^{ij}(t)$ is the realization of $\tau_c^{ij}$ corresponding to the time taken by activating edge $(i, j)$ at time $t$. If node $i$ is chosen for a local computation, which happens with probability $p_i^{\text{comp}}$ then $X^{t+1}(i, w + \tau_l^i(t)) = X^t(i, w)$ for all $w$. Otherwise, $X^{t+1}(j, w) = X^t(j, w)$ for all $w$. At time $t = 0$, $X^0(i, 0) = 1$ and $X^0(i, w) = 0$ for all $w$. Lemma 3 gives a bound on the probability that the time taken by the algorithm to complete $t$ iterations is greater than a given value, depending on variables $X^t$. Note that a Lemma similar to the one by Hendrikx et al. [2019] holds although variable $X$ has been modified.

**Lemma 3.** *We denote $T_{\max}(t)$ the time at which the last node of the system finishes iteration $t$. Then for all $\nu > 0$:*

$$\mathbb{P}\left(T_{\max}(t) \geq \nu t\right) \leq \sum_{w \geq \nu t} \sum_{i=1}^{n} \mathbb{E}\left[X^t(i, w)\right].$$

*Proof.* We first prove by induction on $t$ that for any $i \in \{1, .., n\}$:

$$T_i(t) = \max_{w \in \mathbb{N}, X^t(i, w) > 0} w. \tag{19}$$

To ease notations, we write $w_{\max}(i, t) = \max_{w \in \mathbb{N}, X^t(i, w) > 0} w$. The property is true for $t = 0$ because $T_i(0) = 0$ for all $i$.

We now assume that it is true for some fixed $t > 0$ and we assume that edge $(k, l)$ has been activated at time $t$. For all $i \notin \{k, l\}$, $T_i(t + 1) = T_i(t)$ and for all $w \in \mathbb{N}^*$, $X^{t+1}(i, w) = X^t(i, w)$ so the property is true. Besides, if $j \neq l$,

$$
\begin{aligned}
w_{\max}(k, t + 1) &= \max_{w \in \mathbb{N}^*, X^t(k, w - \tau_c(t)) + X^t(l, w - \tau_c^{kl}(t)) > 0} w \\
&= \max_{w \in \mathbb{N}, X^t(i, w) + X^t(i, w) > 0} w + \tau_c^{kl}(t) \\
&= \tau_c(t) + \max\left(w_{\max}(k, t), w_{\max}(l, t)\right) \\
&= \tau_c^{kl}(t) + \max\left(T_k(t), T_l(t)\right) = T_k(t + 1).
\end{aligned}
$$

Similarly if $k = l$ (a local computation is performed at iteration $t$), then $w_{\max}(k, t + 1) = \tau_l^k(t) + w_{\max}(k, t) = T_k(t) + \tau_l^k(t) = T_k(t + 1)$. Then, we use the union bound and the the fact that having $X^t(i, w) > 0$ is equivalent to having $X^t(i, w) \geq 1$ since $X^t(i, w)$ is integer valued to show that:

$$\mathbb{P}\left(T_{\max}(t) \geq \nu t\right) = \mathbb{P}\left(\max_{w, \sum_{i=1}^{n} X_i^t(w) > 0} w \geq \nu t\right) \leq \mathbb{P}\left(\cup_{w \geq \nu t} \sum_{i=1}^{n} X_i^t(w) \geq 1\right) \leq \sum_{w \geq \nu t} \mathbb{P}\left(\sum_{i=1}^{n} X_i^t(w) \geq 1\right),$$

so using Markov inequality yields:

$$\mathbb{P}\left(T_{\max}(t) \geq \nu t\right) \leq \sum_{w \geq \nu t} \sum_{i=1}^{n} \mathbb{E}\left[X_i^t(w)\right]. \tag{20}$$

$\square$

Variables $X_i^t$ are obtained by linear recursions, so Lemma 3 allows us to bound the growth of variables with a simple recursion formula instead of evaluating a maximum. We write $p_i^{\text{comp}}$ and $p_i^{\text{comm}}$ the probability that node $i$ performs a computation (respectively communication) update at a given time step, and $p_i = p_i^{\text{comp}} + p_i^{\text{comm}}$. We introduce $\underline{p}_{\text{comp}} = \min_i p_i^{\text{comp}}$ and $\bar{p}_{\text{comp}} = \max_i p_i^{\text{comp}}$ (and the same for communication probabilities).

**Lemma 4.** *For all $i$, and all $\nu > 0$, if $\frac{1}{2} \geq \underline{p}_{\text{comp}} = \bar{p}_{\text{comp}} \geq \bar{p}_{\text{comm}}$ then:*

$$\sum_{w \geq (\nu_c + \nu_l)t} \sum_{i=1}^{n} \mathbb{E}\left[X^t(i, w)\right] \to 0 \text{ when } t \to \infty, \tag{21}$$

*with $\nu_c = 6 p_c \tau_c$ and $\nu_l = 9 p_l \tau_l$ where $p_c = 4\bar{p}_{\text{comm}}$ and $p_l = \bar{p}_{\text{comp}}$.*

Note that the constants in front of the $\nu$ parameters are very loose.

*Proof.* Taking the expectation over the edges that can be activated gives, with $\tau_c^{ij}(\tau)$ the probability that $\tau_c^{ij}$ takes value $\tau$ (and the same for $\tau_l$):

$$\mathbb{E}\left[X^{t+1}(i, w)\right] = (1 - p_i) \mathbb{E}\left[X^t(i, w)\right] + p_{\text{comm}} \sum_{j=1}^{n} p_{ij} \sum_{\tau=0}^{\infty} \tau_c^{ij}(\tau) \left(\mathbb{E}\left[X^t(i, w - \tau)\right] + \mathbb{E}\left[X^t(j, w - \tau)\right]\right)$$

$$+ p_i^{\text{comp}} \sum_{\tau=0}^{\infty} \tau_l^{ij}(\tau) \mathbb{E}\left[X^t(i, w - \tau)\right].$$

In particular, for all $i$, $\mathbb{E}\left[X^{t+1}(i, w)\right] \leq \bar{X}^t(w)$ where $\bar{X}^0(w) = 1$ if $w = 0$ and:

$$\bar{X}^{t+1}(w) = (1 - \underline{p}) \bar{X}^t(w) + 2\bar{p}_{\text{comm}} \sum_{\tau=0}^{\infty} \tau_c^{\max}(\tau) \bar{X}^t(w - \tau) + \bar{p}_{\text{comp}} \sum_{\tau=0}^{\infty} \tau_l^{\max}(\tau) \bar{X}^t(w - \tau).$$

with $\tau_c^{\max}(\tau) = \max_{ij} \tau_c^{ij}(\tau)$ (and the same for $\tau_l$). We now introduce $\phi^t(z) = \sum_{w \in \mathbb{N}} z^w \bar{X}^t(w)$. We denote $\phi_c$ and $\phi_l$ the generating functions of $\tau_c^{\max}(\tau)$ and $\tau_l^{\max}(\tau)$. A direct recursion leads to:

$$\phi^t(z) = \left(1 - \underline{p}_{\text{comm}} - \underline{p}_{\text{comp}} + \bar{p}_{\text{comp}} \phi_l(z) + 2\bar{p}_{\text{comm}} \phi_c(z)\right)^t = \left(\phi^1(z)\right)^t.$$

We denote $\phi_{bin}(p, t)$ the generating function associated with the binomial law of parameters $p$ and $t$. With this definition, we have:

$$\phi_{bin}(p_c, t)(\phi_c(z)) \phi_{bin}(p_l, t)(\phi_l(z)) =$$
$$\left[(1 - p_c)(1 - p_l) + (1 - p_c) p_l \phi_l(z) + (1 - p_l) p_c \phi_c(z) + p_c p_l \phi_c(z) \phi_l(z)\right]^t,$$

so we can define:

$$\phi_+^t(z) = (1 + \delta)^t \phi_{bin}(p_c, t)(\phi_c(z)) \phi_{bin}(p_l, t)(\phi_l(z)),$$

where $p_c$, $p_l$ and $\delta$ are such that:

$$\frac{p_c}{1 - p_c} \geq 2\frac{\bar{p}_{\text{comm}}}{1 - \underline{p}}, \quad \frac{p_l}{1 - p_l} = \frac{\bar{p}_{\text{comp}}}{1 - \underline{p}} \text{ and } \delta \geq \frac{1 - \underline{p}}{(1 - p_c)(1 - p_l)} - 1.$$

Since $\bar{p}_{\text{comp}} = \underline{p}_{\text{comp}}$ then $\underline{p} \geq \bar{p}_{\text{comp}}$. Therefore, these conditions are satisfied for $p_c$ and $p_l$ as given by Lemma 4 and $\delta = (1 - p_c)^{-1} - 1$. Then $(1 + \delta)(1 - p_c)(1 - p_l) \geq 1 - \underline{p}$, $(1 + \delta)(1 - p_c) p_l \geq \bar{p}_{\text{comp}}$ and $(1 + \delta)(1 - p_l) p_c \geq 2\bar{p}_{\text{comm}}$. This means that if we write $\phi^1(z) = a_0 + a_c \phi_c(z) + a_l \phi_l(z)$ and $\phi_+^1(z) = b_0 + b_c \phi_c(z) + b_l \phi_l(z)$ then $b_0 \geq a_0$, $b_c \geq a_c$ and $b_l \geq a_l$. In particular, all the coefficients

of $\phi^t$ are smaller than the coefficients of $\phi_+^t$ where both functions are integral series. Therefore, if we call $Z_t$ the random variables associated with the generating function $(1+\delta)^{-t}\phi_+^t$ then for all $i, t, w$:

$$\mathbb{E}\left[X^t(i,w)\right] \leq (1+\delta)^t \mathbb{P}\left(Z_t = w\right), \tag{22}$$

where $Z_t = Z_c^t + Z_l^t = Bin(p_c, t)(Z_c) + Bin(p_l, Z_l)(\tau_l)$ where $Z_c$ and $Z_l$ are the random variables modeling the time of one communication or computation update. We can then use the bound $p(Z_t \geq (\nu_c + \nu_l)t) \leq p(Z_c^t \geq \nu_c t) + p(Z_l^t \geq \nu_l t)$. This way, we can bound the *communication* and *computation* costs independently. Then, we write a Chernoff bound, i.e. for any $\lambda > 0$:

$$\mathbb{P}\left(Z_c^t \geq \nu t\right) \leq e^{-\lambda \nu t} \mathbb{E}\left[e^{\lambda Z_c^t}\right] = e^{-\lambda \nu t} \mathbb{E}\left[e^{\lambda Z_c}\right]^t = e^{-\lambda \nu t} \left[1 - p_c + p_c \sum_{\tau=0}^{\infty} p_c(\tau) e^{\lambda \tau}\right]^t,$$

where $S_c$ is the sum of $t$ i.i.d. random variables drawn from $\tau_c$. If $Z_c = \tau_c$ with probability 1 (deterministic delays) then this reduces to:

$$\mathbb{P}\left(Z_c^t \geq \nu_c t\right) \leq e^{-\lambda \nu_c t}\left[1 - p_c + p_c e^{\lambda \tau_c}\right].$$

Finally, we take $\nu_c = k p_c \tau_c$, $\lambda = \frac{1}{\tau_c}\ln(k)$ and we use the basic inequality $\ln(1+x) \geq \frac{x}{1+x}$ to show that:

$$-\ln\left[\mathbb{P}\left(Z_c^t \geq \nu_c t\right)\right] \geq t\left[\lambda \nu_c - p_c\left(e^{\lambda \tau_c} - 1\right)\right] \geq t(k(\ln(k)-1)-1)p_c.$$

Using the same log inequality and the fact that $p_c \geq \frac{1}{2}$ yields:

$$\ln\left(1+\delta\right) = -\ln(1-p_c) \leq \frac{p_c}{1-p_c} \leq 2p_c.$$

Therefore, choosing $k = 6$ ensures that $k(\ln(k)-1)-1 \geq 3$ and so:

$$(1+\delta)^t \mathbb{P}\left(Z_c^t \geq \nu_c t\right) \leq e^{-t p_c}.$$

We can apply the same reasoning to $Z_l^t$, and the bound is still valid with $k=9$ because $p_l = \bar{p}_{\text{comp}} \geq \bar{p}_{\text{comm}} = p_c/4$. We finish the proof by using Equation (22). □

## D  Algorithm Performances

ADFS has a linear convergence rate because it results from using generalized APCG. Yet, it is not straightforward to derive hyperparameters that lead to a rate that is fast and that can be easily interpreted. The goal of this section is to choose such parameters when the functions $f_{i,j}$ are smooth.

### D.1  Smallest eigenvalue of the Laplacian of the augmented graph

The strong convexity of $q_A$ on the orthogonal of the kernel of $A$ is equal to $\sigma_A = \lambda_{\min}^+\left(A^T\Sigma^{-1}A\right)$, the smallest non-zero eigenvalue of $A^T\Sigma^{-1}A$. Indeed, $\Sigma^{-1}$ is a diagonal matrix with strictly positive entries only so $\text{Ker}(A^T\Sigma^{-1}A) = \text{Ker}(A)$. The goal of this section is to prove that for a meaningful choice of $\mu$, the smallest eigenvalue of the Laplacian of the augmented graph is not too small compared to the Laplacian of the actual graph. More specifically, we prove the following result:

**Lemma 5.** *If for all virtual edge between a node $i$ and its virtual node $j$, $\mu_{ij}$ is such that $\mu_{ij}^2 = \frac{\lambda_{\min}^+(L)}{\sigma_i \kappa_i} L_{i,j}$, then for any $x \in \mathbb{R}^{E+nm}$*

$$\|x\|_{A^T\Sigma^{-1}A}^2 \geq \min_{i \in \{1,\dots,n\}} \frac{\lambda_{\min}^+(L)}{2\sigma_i \kappa_i} \|x\|_{A^\dagger A}^2.$$

*In particular, $\sigma_A \geq \min \frac{\lambda_{\min}^+(L)}{2(\sigma_i + \sum_{j=1}^m L_{ij})}$.*

We prove this Lemma in three steps. Lemma 6 gives a characterization of the eigenvalues of $A^T\Sigma^{-1}A$ in terms of a determinant equation. Then, Lemma 7 a condition on the coefficients of a matrix so that solutions to the previous equations exist. Finally, Lemma 8 transforms this condition into the lower bound of Lemma 5.

**Lemma 6.** *We denote $L$ the Laplacian matrix of the communication graph and define $\alpha_i = \frac{\mu_{ij}^2}{L_{ij}}$ and $\Delta_\lambda$ the diagonal matrix such that $(\Delta_\lambda)_{ii} = \lambda \left( \sigma_i + \frac{\alpha_i S_i}{\alpha_i - \lambda} \right)$ where $S_i = \sum_{j=1}^m L_{i,j}$.*
*and . Then, the eigenvalues of $A^T \Sigma^{-1} A$, noted $\mathrm{eig}(A^T \Sigma^{-1} A)$ are such that:*

$$\mathrm{eig}(A^T \Sigma^{-1} A) \subset \{\lambda, \ \det(L - \Delta_\lambda) = 0\} \cup \{\alpha_1, ..., \alpha_n\}.$$

*Proof.* For any rectangular matrix $Q$, all non-zero singular values of the matrix $Q^T Q$ are also non-zero singular values of the matrix $Q Q^T$, so we can analyze the spectrum of the matrix $\tilde{L} = \Sigma^{-1/2} A A^T \Sigma^{-1/2}$ instead of the spectrum of $A^T \Sigma^{-1} A$. Then, we denote $\mu_{ij}^2$ the weight of the *virtual edge* $(i, j)$ and $M$ the diagonal matrix of size $nm$ which is such that ${e^{(i,j)}}^T M e^{(i,j)} = \mu_{ij}^2$ for all virtual nodes. $M_{n,m}$ is the matrix of size $n \times nm$ such that $(M_{n,m} e_{ij})^{(i)} = \mu_{ij}^2$ for all virtual edges $(i, j)$ and all other entries are equal to 0. Finally, $\tilde{S}$ is the diagonal matrix of size $n$ such that $(\tilde{S})_{ii} = \sum_{j=1}^n \mu_{ij}^2$. All *communication nodes* are linked by the true graph, whereas all *virtual nodes* are linked to their corresponding communication node. Then, if we denote $L$ the Laplacian matrix of the original true graph, the rescaled Laplacian matrix of the augmented graph writes:

$$\tilde{L} = \Sigma^{-1/2} \begin{pmatrix} L + \tilde{S} & -M_{n,m} \\ -M_{n,m}^T & M \end{pmatrix} \Sigma^{-1/2}. \tag{23}$$

Therefore, if we split $\Sigma$ into two diagonal blocks $\Sigma_{\mathrm{comm}}$ (for the communication nodes) and $\Sigma_{\mathrm{comp}}$ (for the computation nodes) and apply the block determinant formula, we obtain:

$$\begin{aligned}
\det(\tilde{L} - \lambda I_d) = {} & \det(\Sigma_{\mathrm{comp}}^{-1} M - \lambda Id) \\
& \times \det(\Sigma_{\mathrm{comm}}^{-1/2} L \Sigma_{\mathrm{comm}}^{-1/2} + \Sigma_{\mathrm{comm}}^{-1} \tilde{S} - \lambda Id - \\
& \Sigma_{\mathrm{comm}}^{-\frac{1}{2}} M_{n,m} \Sigma_{\mathrm{comp}}^{-\frac{1}{2}} \left( \Sigma_{\mathrm{comp}}^{-1} M - \lambda Id \right)^{-1} \Sigma_{\mathrm{comp}}^{-\frac{1}{2}} M_{n,m}^T \Sigma_{\mathrm{comm}}^{-\frac{1}{2}} ).
\end{aligned}$$

Then, we choose $M$ such that $\Sigma_{\mathrm{comp}}^{-1} M = diag(\alpha_1, ..., \alpha_n)$, meaning that $\mu_{ij}^2 = \alpha_i L_{i,j}$. With this choice, the eigenvalues of $\tilde{L}$ are either included in $\{\alpha_1, ..., \alpha_n\}$ or the solutions of:

$$\det(\Sigma^{-\frac{1}{2}} (L - \Delta_\lambda) \Sigma^{-\frac{1}{2}}) = \det(\Sigma^{-\frac{1}{2}})^2 \det(L - \Delta_\lambda) = 0, \tag{24}$$

where $\Delta_\lambda$ is the diagonal matrix such that:

$$(\Delta_\lambda)_{ii} = \sigma_i \lambda + \sum_{j=1}^n \mu_{ij}^2 \left( \frac{\mu_{ij}^2}{\mu_{ij}^2 - L_{i,j} \lambda} - 1 \right). \tag{25}$$

We finish the proof of Lemma 6 by taking $\mu_{ij}^2 = \alpha_i L_{i,j}$ and by remarking that $\det(\Sigma^{-\frac{1}{2}}) \neq 0$. $\square$

We now need to understand $\mathrm{Ker}(L - \Delta_\lambda)$ more in details in order to have a lower bound on $\lambda$. We deduce bounds from the following lemma:

**Lemma 7.** *The following relations hold:*

- $\|x\|_{A^T \Sigma^{-1} A}^2 \geq \lambda_{\min}^+(A^T \Sigma^{-1} A) \|x\|_{A^\dagger A}^2$ *for any $x \in \mathbb{R}^E$.*
- *If $\lambda > 0$ is such that $\mathrm{Ker}(L - \Delta_\lambda) \neq \{0\}$ then $\lambda \geq \min_i \alpha_i$ or $\max_i (\Delta_\lambda)_{ii} \geq \lambda_{\min}^+(L)$.*

*Proof.* We know that $\mathrm{Ker}(\Sigma^{-1}) = \{0\}$ since all local functions are strongly convex and smooth. Therefore, $\mathrm{Ker}(A^T \Sigma^{-1} A) = \mathrm{Ker}(A)$ and the first point follows directly.

We now fix $\lambda$ such that $0 < \lambda < \min_i \alpha_i$, meaning that $(\Delta_\lambda)_{ii} > 0$ for all $i$. Let $x \in \mathrm{Ker}(L - \Delta_\lambda)$. We write $x$ as $x = x_+ + \bar{x} \mathbb{1}$, with $\bar{x} = \frac{1}{n} \sum_{i=1}^n x_i$ and $x_+$ such that $\mathbb{1}^T x_+ = 0$ where $\mathbb{1}$ is the vector with all entries equal to 1. Then the following relation holds:

$$0 = (L - \Delta_\lambda) x = L x_+ - \Delta_\lambda x_+ - \bar{x} \Delta_\lambda \mathbb{1}.$$

Therefore,

$$\mathbb{1}^T (L - \Delta_\lambda) x = 0 = -\mathbb{1}^T \Delta_\lambda x_+ - \bar{x} \mathbb{1}^T \Delta_\lambda \mathbb{1},$$

which can be rewritten

$$\bar{x} = -\frac{\mathbb{1}^T \Delta_\lambda x_+}{\mathrm{Tr}(\Delta_\lambda)}. \tag{26}$$

In the same way, we obtain:

$$x_+^T(L - \Delta_\lambda)x = 0 = x_+^T L x_+ - x_+^T \Delta_\lambda x_+ - \bar{x} x_+^T \Delta_\lambda \mathbb{1},$$

and replacing $\bar{x}$ by its expression yields:

$$x_+^T L x_+ = x_+^T \Delta_\lambda x_+ - \frac{(x_+^T \Delta_\lambda \mathbb{1})^2}{\mathrm{Tr}(\Delta_\lambda)}.$$

Using that $\mathrm{Tr}(\Delta_\lambda) > 0$, we deduce that:

$$\lambda_{\min}^+(L)\|x_+\|^2 \le x_+^T L x_+ = x_+^T \Delta_\lambda x_+ - \frac{(x_+^T \Delta_\lambda \mathbb{1})^2}{\mathrm{Tr}(\Delta_\lambda)} \le \|x_+\|^2 \max_i(\Delta_\lambda)_{ii}. \tag{27}$$

If we assume that $\max_i(\Delta_\lambda)_{ii} < \lambda_{\min}^+(L)$ then Equation (27) implies that $x_+ = 0$ since all the entries of $\Delta_\lambda$ are positive. We then deduce from Equation (26) that $\bar{x} = 0$ as well in this case, which implies that if $x \in \mathrm{Ker}(L - \Delta_\lambda)$ then $x = 0$. Therefore, $\mathrm{Ker}(L - \Delta_\lambda) = \{0\}$ if $\max_i(\Delta_\lambda)_{ii} < \lambda_{\min}^+(L)$ and $0 < \lambda < \min_i \alpha_i$, which finishes the proof. $\qquad\square$

**Lemma 8.** *If we choose* $\alpha_i = \frac{\lambda_{\min}^+(L)}{\sigma_i \kappa_i}$, *and* $\lambda < \min_i \frac{\alpha_i}{2}$ *then* $\max_i(\Delta_\lambda)_{ii} < \lambda_{\min}^+(L)$

*Proof.* The function $\lambda \mapsto \max_i(\Delta_\lambda)_{ii}$ is strictly increasing in $\lambda$ on the interval $[0, \min_i \alpha_i[$. Therefore, we simply need to solve the following equation on this interval:

$$\lambda\left(\sigma_i + \frac{\alpha_i S_i}{\alpha_i - \lambda}\right) = \lambda_{\min}^+(L). \tag{28}$$

This equation can be rewritten:

$$\lambda^2 - \lambda\left(\alpha_i \kappa_i + \frac{\lambda_{\min}^+(L)}{\sigma_i}\right) + \frac{\alpha_i}{\sigma_i}\lambda_{\min}^+(L) = 0.$$

We choose $\alpha_i = \frac{\lambda_{\min}^+(L)}{\sigma_i \kappa_i}$ to get:

$$\lambda^2 - 2\lambda\frac{\lambda_{\min}^+(L)}{\sigma_i} + \frac{1}{\kappa_i}\left(\frac{\lambda_{\min}^+(L)}{\sigma_i}\right)^2 = 0.$$

This is a second degree equation and its smallest root is given by:

$$\lambda_i^- \ge \frac{\lambda_{\min}^+(L)}{\sigma_i}\left(1 - \sqrt{1 - \frac{1}{\kappa_i}}\right).$$

In particular, all its roots are bigger than the smallest one, so using the inequality $\sqrt{1-x} \le 1 - \frac{x}{2}$, we obtain that all the solutions $\lambda_i$ of Equation (28) must satisfy:

$$\lambda_i \ge \frac{\lambda_{\min}^+(L)}{2\sigma_i \kappa_i} = \frac{\alpha_i}{2}.$$

We finish the proof by using the monotonicity of the function $\lambda \mapsto \max_i(\Delta_\lambda)_{ii}$. $\qquad\square$

### D.2 Eigengap of the augmented graph

This section aims as justifying the $\tilde{\gamma}$ notation. Recall that it is defined such that $\tilde{\gamma} = \min_{(k,\ell) \in E^{\mathrm{comm}}} \frac{\lambda_{\min}^+(L)n^2}{\mu_{k\ell}^2 e_{k\ell}^T A^\dagger A e_{k\ell} E^2}$. We show in this section that for any given family of regular graphs, there exists a constant $C_\gamma$ independent of the size of the graph such that $C_\gamma \tilde{\gamma} \ge \gamma$. Matrix $A$ depends on $\mu$, and we consider in this section that $\mu_{k\ell}^2 = \mu_0$ for all communication edges $(k, \ell)$. Similar results can be obtained when $\mu$ is heterogeneous.

**Regular graphs.** We say that a family of graph is regular if there exists $C_\gamma > 0$ such that $e_{k\ell}^T A^\dagger A e_{k\ell} \leq C_\gamma \frac{n}{E}$ for any $n > 2$.

Recall that $E$ is the number of edges (usually constrained by the graph family and the number of nodes), and $e_{k\ell}^T A^\dagger A e_{k\ell}$ is the effective resistance of edge $(k, \ell)$. This assumption seems a bit technical but it simply requires that all edges contribute equally to the connectivity of the graph, and therefore is related to how symmetric the graph is. In particular, it is verified with $C_\gamma = 1$ for any completely symmetric graph, such as the complete graph or the ring. Since $e_{k\ell}^T A^\dagger A e_{k\ell} \leq 1$, it is also satisfied any time the ratio $n/E$ is bounded below, and in particular for the grid, the hypercube, or any graph with bounded degree. Under these assumptions, and for any communication edge $(k, \ell)$:

$$\frac{\lambda_{\min}^+(L)n^2}{\mu_{k\ell}^2 e_{k\ell}^T A^\dagger A e_{k\ell} E^2} \geq \frac{\gamma}{C_\gamma} \frac{\lambda_{\max}(L)n}{\mu_{k\ell}^2 E} \geq \frac{\gamma}{C_\gamma} \frac{\text{Trace}(L)}{\mu_0^2 E} = 2\frac{\gamma}{C_\gamma}.$$

Here, we used the fact that $\text{Trace}(L) = 2\mu_0^2 E$, which can be deduced directly from the form of $A$ (each edge has weight $\mu_0^2$ and contributes two times, one for each end). We conclude by using the fact that since the previous inequalities are true for any $(k, \ell) \in E^{\text{comm}}$, it is in particular true for $\tilde{\gamma}$.

### D.3 Communication rate and local rate

We know that the rate of ADFS can be written as the minimum of a given quantity over all edges of the graph. This quantity will be very different whether we consider communication edges or virtual edges. In this section, we give lower bounds for each type of edge, and show that we can trade one for the other by adjusting the probability of communication.

**Lemma 9.** *With the choice of parameters of Theorem 3, parameter $\rho$ satisfies:*

$$\rho \geq \frac{1}{\sqrt{2}n} \min\left( p_{\text{comm}} \Delta_p \sqrt{\frac{\tilde{\gamma}}{2\kappa}}, p_{\text{comp}} \frac{\sqrt{r_\kappa}}{S_{\text{comp}}} \right). \tag{29}$$

*Proof.* Recall that the rate $\rho$ is defined as:

$$\rho^2 = \min_{k\ell} \frac{p_{k\ell}^2}{\mu_{k\ell}^2 e_{k\ell}^T A^\dagger A e_{k\ell}} \frac{\lambda_{\min}^+(\tilde{L})}{\sigma_k^{-1} + \sigma_\ell^{-1}}, \tag{30}$$

and that Lemma 5 ensures that

$$\lambda_{\min}^+(\tilde{L}) \geq \frac{\lambda_{\min}^+(L)}{2\sigma\kappa}.$$

Therefore, for communication edges the rate writes:

$$\rho_{\text{comm}}^2 \geq \min_{(k,\ell) \in E^{\text{comm}}} \left( \frac{1}{\sigma_k} + \frac{1}{\sigma_\ell} \right)^{-1} \frac{p_{k\ell}^2}{\mu_{k\ell}^2 e_{k\ell}^T A^\dagger A e_{k\ell}} \frac{\lambda_{\min}^+(L)}{2\sigma\kappa}. \tag{31}$$

If we take $\sigma_k = \sigma$ for all $k$ and $p_{k\ell}^2 \geq \Delta_p^2 p_{\text{comm}}^2 / |E|^2$ (corresponding to a homogeneous case), then we can make $\tilde{\gamma}$ appear to obtain:

$$\rho_{\text{comm}}^2 \geq \Delta_p^2 \frac{\tilde{\gamma}}{\kappa} \frac{p_{\text{comm}}^2}{4n^2}. \tag{32}$$

For "computation edges", we can write:

$$\rho_{\text{comp}}^2 \geq \min_{ij} \frac{p_{ij}^2}{2\left(\sigma_i^{-1} + L_{i,j}^{-1}\right)} \frac{\sigma\kappa_i}{\lambda_{\min}^+(L) L_{i,j}} \frac{\lambda_{\min}^+(L)}{\sigma\kappa}, \tag{33}$$

because $e_{ij}^T A^\dagger A e_{ij} = 1$ when $(i, j)$ is a virtual edge (because it is part of no cycle). Since $S_{\text{comp}} = \frac{1}{n} \sum_{i=1}^n \sum_{j=1}^m \sqrt{1 + L_{i,j}\sigma_i^{-1}}$, this can be rewritten:

$$\rho_{\text{comp}}^2 \geq \frac{r_\kappa}{2} \frac{p_{\text{comp}}^2}{n^2 S_{\text{comp}}^2}. \tag{34}$$

$\square$

## D.4 Execution time

Now that we have specified the rate of ADFS (improvement per iteration), we can bound the time needed to reach precision $\varepsilon$ by plugging in the expected time to execute the schedule. In particular, we show in this section Theorem 6, which is a more precise version of Theorem 3.

We introduce $\Delta_p$, $r_\kappa$ and $c_\tau$ to quantify how heterogeneous the system is. More specifically, we can define $\sigma = \max_i \sigma_i$, $\kappa_i = 1 + \sigma_i^{-1} \sum_{j=1}^m L_{i,j}$ and $\kappa_s = \max_i \kappa_i$. Since they are not all equal, we introduce $r_\kappa = \min_i \kappa_i / \kappa_s$. We choose the probabilities of virtual edges, such that $\sum_{j=1}^m p_{ij}$ is constant for all $i$ and such that $p_{ij} = p_{\text{comp}}(1 + L_{i,j}\sigma_i^{-1})^{\frac{1}{2}}/(nS_{\text{comp}})$ for $S_{\text{comp}} = n^{-1} \sum_{i=1}^n \sum_{j=1}^m (1 + L_{i,j}\sigma_i^{-1})^{\frac{1}{2}}$. When $(k, \ell)$ is a communication edge, we further assume that $p_{k\ell} \geq \Delta_{\text{p}} p_{\text{comm}}/|E|$ for some constant $\Delta_{\text{p}} \leq 1$ and $p_{\text{comm}}^{\max} \leq c_\tau p_{\text{comm}}$ for some $c_\tau > 0$.

**Theorem 6.** *We choose $\mu_{k\ell}^2 = \frac{1}{2}$ for communication edges, $\mu_{ij}^2 = \frac{\lambda_{\min}^+(L)}{\sigma \kappa_i} L_{i,j}$ for computation edges and $p_{\text{comm}} = \min\left(\frac{1}{2}, \left(1 + \sqrt{\frac{\tilde{\gamma}}{\kappa_{min}}} S_{\text{comp}}\right)^{-1}\right)$. Then, running Algorithm 1 for $K = \rho^{-1} \log\left(\varepsilon^{-1}\right)$ iterations guarantees $\mathbb{E}\left[\|\theta_K - \theta^\star\|^2\right] \leq C_0 \varepsilon$, and takes time $T(K)$, with $T(K)$ bounded by:*

$$T(K) \leq 2C \left(\frac{m + \sqrt{m\kappa_s}}{\sqrt{2r_\kappa}} + \frac{(1 + 4c_\tau\tau)}{\Delta_{\text{p}}}\sqrt{\frac{\kappa_s}{\tilde{\gamma}}}\right) \log\left(\frac{1}{\varepsilon}\right)$$

*with probability tending to 1 as $\rho^{-1} \log\left(\varepsilon^{-1}\right) \to \infty$, where $C$ is the same as in Theorem 2.*

In heterogeneous settings, $\sigma_i$ and sampling probabilities may be adapted to recover good guarantees, but this is beyond the scope of this paper. Note that taking computing probabilities exactly equal for all nodes is not necessary to ensure convergence, and only slightly slows down convergence. Indeed, it is always possible to analyze a schedule for which all nodes have exactly the same probability of local update by adding a probability of doing nothing for time 1 as a local update to the nodes that are chosen less frequently. If we denote $p_{\text{wait}}$ the probability that we need to add so that all nodes have the same probability of being selected, then $p_{\text{comp}} + p_{\text{comm}} = 1 - p_{\text{wait}}$ so $\theta_{\text{comp}}$ will be slightly smaller for a given $p_{\text{comm}}$. The actual algorithm can only be faster so this just gives a rough upper bound on the time to convergence.

*Proof.* Using Theorem 2 on the average time per iteration, we know that as long as $p_{\text{comp}} > p_{\text{comm}}$, the execution time of the algorithm verifies the following bound for some $C > 0$ with high probability:

$$T(K) \leq \frac{C}{n}\left(p_{\text{comp}} + 2\tau p_{\text{comm}}^{\max}\right)K \tag{35}$$

Algorithm 1 requires $-\log(1/\varepsilon)/\log(1 - \rho)$ iterations to reach error $\varepsilon$. Using that $\log(1 + x) \leq x$ for any $x > -1$, we get that using $K_\varepsilon = \log(1/\varepsilon)\rho^{-1}$ instead also guarantees to make error less than $\varepsilon$. We now optimize the bound in $\rho$:

$$\frac{T(K_\varepsilon)}{\log\left(\varepsilon^{-1}\right)} \leq \frac{C}{n\rho}\left(p_{\text{comp}} + 2\tau p_{\text{comm}}^{\max}\right) \tag{36}$$

If we rewrite this in terms of $\rho_{\text{comm}}$ and $\rho_{\text{comp}}$, we obtain:

$$\frac{T(K_\varepsilon)}{\log\left(\varepsilon^{-1}\right)} \leq C \max\left(T_1(p_{\text{comm}}), T_2(p_{\text{comm}})\right) \tag{37}$$

with

$$T_1(p_{\text{comm}}) = \frac{1}{n\rho_{\text{comm}}}\left(p_{\text{comp}} + 2c_\tau\tau p_{\text{comm}}\right) = \frac{2}{\Delta_p}\left(2c_\tau\tau - 1 + \frac{1}{p_{\text{comm}}}\right)\sqrt{\frac{\kappa}{\tilde{\gamma}}} \tag{38}$$

and

$$T_2(p_{\text{comm}}) = S_{\text{comp}}\sqrt{\frac{2}{r_\kappa}}\left(\frac{1 + (2c_\tau\tau - 1)p_{\text{comm}}}{1 - p_{\text{comm}}}\right) = S_{\text{comp}}\sqrt{\frac{2}{r_\kappa}}\left(1 + 2\tau\frac{p_{\text{comm}}}{1 - p_{\text{comm}}}\right) \tag{39}$$

$T_1$ is a continuous decreasing function of $p_{\text{comm}}$ with $T_1 \to \infty$ when $p_{\text{comm}} \to 0$. Similarly, $T_2$ is a continuous increasing function of $p_{\text{comm}}$ such that $p_{\text{comm}} \to \infty$ when $p_{\text{comm}} \to 1$. Therefore,

the best upper bound on the execution time is given by taking $p_{\text{comm}} = p^*$ where $p^*$ is such that $T_1(p^*) = T_2(p^*)$ and so $\rho_{\text{comm}}(p^*) = \rho_{\text{comp}}(p^*)$.

$$\frac{T(K_\varepsilon)}{\log(\varepsilon^{-1})} \leq CT_1(p^*) \tag{40}$$

Then, $p^*$ can be found by finding the root in $]0,1[$ of a second degree polynomial. In particular, $p^*$ is the solution of:

$$p_{\text{comp}}^2 = p_{\text{comm}}^2 \frac{\tilde{\gamma}\Delta_p^2}{2\kappa r_\kappa} S_{\text{comp}}^2 = (1 - p_{\text{comm}})^2 \tag{41}$$

which leads to $p^* = \left(1 + \sqrt{\frac{\tilde{\gamma}}{2\kappa_{\min}}}\Delta_p S_{\text{comp}}\right)^{-1}$.

$$\frac{T(K_\varepsilon)}{\log(\varepsilon^{-1})} \leq 2\frac{C}{\Delta_p}\left(2c_\tau\tau - 1 + \frac{1}{p^*}\right)\sqrt{\frac{\kappa}{\tilde{\gamma}}}$$

$$\leq 2C\left(2\tau\frac{c_\tau}{\Delta_p}\sqrt{\frac{\kappa}{\tilde{\gamma}}} + \frac{1}{\sqrt{2r_\kappa}}S_{\text{comp}}\right)$$

Finally, we use the concavity of the square root to show that:

$$S_{\text{comp}} = \frac{1}{n}\sum_{i=1}^{n}\sum_{j=1}^{m}\sqrt{1 + L_{i,j}\sigma_i^{-1}}$$

$$\leq \frac{1}{n}\sum_{i=1}^{n}m\sqrt{\sum_{j=1}^{m}\frac{1}{m}\left(1 + L_{i,j}\sigma_i^{-1}\right)}$$

$$\leq \frac{1}{n}\sum_{i=1}^{n}m\sqrt{1 + \frac{1}{m}(\kappa_i - 1)}$$

$$\leq m + \sqrt{m\kappa}$$

Yet, this analysis only works as long as $p^* \leq 1/2$. When this constraint is not respected, we know that: $\tilde{\gamma}S_{\text{comp}}^2 \leq 2\kappa r_\kappa$. In this case, we can simply choose $p_{\text{comp}} = p_{\text{comm}} = \frac{1}{2}$ and then $\rho_{\text{comm}} \leq \rho_{\text{comp}}$, so

$$\frac{T(K_\varepsilon)}{\log(\varepsilon^{-1})} \leq CT_1\left(\frac{1}{2}\right) = 2\frac{C}{\Delta_p}(1 + 2c_\tau\tau)\sqrt{\frac{\kappa}{\tilde{\gamma}}} \tag{42}$$

The sum of the two bounds is a valid upper bound in all situations, which finishes the proof. □

## E  Experimental setting

### E.1  Experimental Setting

We detail in this section the exact experimental setting in which simulations were made. All algorithms used out-of-the-box parameters given by theory. Batch algorithms as well as ESDACD were given the exact $\kappa_b$. The datasets we used are the first million samples of the Higgs dataset (11 million samples and 28 attributes) and the Covtype.binary.scale dataset (581,012 samples and 54 attributes). Both datasets are available at https://www.csie.ntu.edu.tw/~cjlin/libsvmtools/datasets/binary.html. To obtain the local dataset $X_i \in \mathbb{R}^{m \times d}$ of each node, we drew $m$ samples at random from the base dataset, so that datasets of different nodes may overlap. We used the logistic loss with quadratic regularization, meaning that the function at node $i$ is:

$$f_i : \theta_i \mapsto \sum_{j=1}^{m}\log\left(1 + \exp(-l_{i,j}X_{i,j}^T\theta_i)\right) + \frac{\sigma_i}{2}\|\theta_i\|^2,$$

where $l_{i,j} \in \{-1, 1\}$ is the label associated with $X_{i,j}$, the $k$-th sample of node $i$. We chose $m = 10^4$ and $\sigma = 1$ for all simulations. Note that local functions are not normalized (not divided by $m$) so this

(a) Higgs, $\tau = 5$　　　　(b) Covtype, $\tau = 5$　　　　(c) Higgs, $\tau = 50$

Figure 4: Simulations on the logistic regression task with $m = 10^4$ points per node, regularization parameter $\sigma = 1$ on grid networks of size 100.

actually corresponds to a regularization value of $\sigma_i = 10^{-4}$ with usual formulations. Computation delays were chosen constant equal to 1 and communication delays constant equal to 5.

As said in the main text, plots are shown for *idealized times* in order to abstract implementation details as well as ensure that reported timings were not impacted by the cluster status (available bandwidth for example). Note that for ADFS, nodes perform the schedule described in Section 4 and are considered free to start the next iteration as soon as they send their a gradient as long as they already received the neighbor's gradient (non-blocking send). Note that although Algorithm 1 returns vector $\Sigma^{-1}v_t$ to computeAn Accelerated Decentralized Stochastic Proximal Algorithm for Finite Sums the error, we used the vector $\Sigma^{-1}y_t$ instead. Both have similar asymptotic convergence rates but the error was more stable using $\Sigma^{-1}y_t$. The error that we plot is the average error over all nodes at a given time. More specifically, all nodes compute the error at specific iteration number as $F(\Sigma^{-1}y_t)$. Then, we average all these errors and the time reported is the time at which the last node finishes this iteration. We set the parameters $\mu$ and and $p_{ij}$ as described in Section 5. Yet, Taking slightly lower values for $\theta$ (for example dividing it by 2) seems to help in some cases, pr

Similarly to Table 1, we assume that computing the dual gradient of a function $f_i$ is as long as computing $m$ proximal operators of $f_{i,j}$ functions. This greatly benefits to MSDA and ESDACD since in the case of logistic regression, the proximal operator for one sample has no analytic solution but can be efficiently computed as the result of a one-dimensional optimization problem [Shalev-Shwartz and Zhang, 2013]. The inner problem corresponding to computing $\nabla f_i^*$ was solved by performing 500 steps of accelerated gradient descent. For Point-SAGA, ADFS and DSBA, 1D prox were computed using 5 steps of Newton's method (in one dimension). Both used warm-starts, *i.e.* the initial parameter for these inner problems was the solution for the last time the problem was solved. The step-size $\alpha$ of DSBA was chosen as $1/(4L_{\max})$ instead of $1/(24L_{\max})$ where $L_{\max} = \max_{i,j} L_{i,j}$. DSBA was unstable for larger values of $\alpha$.

### E.2 Centralized Algorithms

In this section, we perform a quick comparison with the centralized algorithm Katyusha [Allen-Zhu, 2017]. We assume that the allreduce communication steps take time $\Delta$ where $\Delta$ is the diameter of the graph. We implement the mini-batch version of this algorithm and set the mini-batch size so that $b = 1 + \Delta\tau$, *i.e.*, the algorithm spends as much time computing as communicating (not counting the full gradient steps). Counting computation time in terms of effective passes over the dataset is slightly unfair to Katyusha that has a cheaper per-example cost. Yet, this is only a (small) constant factor in the case of logistic regression.

We observe on Figures 4a and 4b that Katyusha and ADFS have comparable rates when $\tau = 5$. This is mainly due to the fact that communications are quite fast so the effective mini-batch size is 9100 in this case (diameter of the graph is 18 so 91 per node), which is quite small compared to the $10^6$ total samples. Figure 4c shows that ADFS can outperform Katyusha on the Higgs dataset (on which it was slower when taking $\tau = 5$) when delays are big ($\tau = 50$). Indeed, the effective batch size in this case is 91000, which is about 10% of the dataset and so Katyusha does not take full advantage of the stochastic optimization speedup. Yet, it is still significantly faster than MSDA. Note that in the case of $\tau = 50$, we set $p_{\text{comm}}$ such that $\tau p_{\text{comm}} = p_{\text{comp}}$. This choice slightly reduced the number communications and led to a faster algorithm by reducing communication time and synchronization barriers. Overall, we see that, contrary to existing decentralized methods, ADFS can be competitive

with a distributed implementation of Katyusha, especially when delays are high. Yet, a more detailed study reporting actual computing times with fully optimized implementations would be needed to compare the algorithms further. Indeed, some simulation choices favored ADFS (normalized time, neglecting overhead induced by the prox), whereas other favored Katyusha (constant delays, homogeneous setting).

More fundamentally, Katyusha and ADFS are based on two distinct distribution paradigms. On the one hand, centralized algorithms use less noisy gradients because they have an effective mini-batch size of at least $n$. This grants them linear speedup given that the batch size is small enough. Yet, the batch size usually has to be quite high because it needs to grow linearly with the communication time and the diameter of the graph in order to avoid spending more time communicating than computing so centralized approaches are not necessarily the best option on high-latency networks. On the other hand, decentralized algorithms such as ADFS can work with very small batches but they do not get the mini-batch noise reduction from computing on $n$ nodes in parallel the way Katyusha does. Indeed, similarly to "Local-SGD" [Lin et al., 2018, Patel and Dieuleveut, 2019] approaches, each node locally runs an accelerated variance-reduced algorithm. This confirms that decentralized algorithms, and in particular ADFS, can be well-suited for distributed stochastic optimization with delays.

### E.3 Code

A Python implementation of ADFS can be found on GitHub (`https://github.com/HadrienHx/ADFS_NeurIPS`). The goal of this code is to show how to implement ADFS and encourage its use as a baseline. The code implements ADFS to solve the Logistic Regression problem on a 2D grid. It generates a synthetic binary classification dataset. Our implementation leverages the fact that Logistic Regression is a linear model to only store 2 scalars per virtual node instead of 2 full vectors, thus showing how to use sparse updates. The code is not optimized and not intended to be particularly fast, but rather to show how to go from the pseudo-code in Algorithm 1 to an actual implementation.