[Reviews · NeurIPS 2019]

Reviewer 1



MINOR COMMENTS: Page 3, line 85. "low communication without sparsity assumptions". Please add references on such results. Page 3, line 99. "much faster than computing the gradient of the conjugate[..]". I agree with this, but it also shows that the comparison is not quite fair, since e.g. MSDA was not designed for finite sums on each node. Page 5, line 165. "generalized version of the APCG algorithm". I believe it is worth at this point to be more precise on what generalizations have been included in this algorithm, and why they matter.

Reviewer 2



This paper proposes a new efficient accelerated decentralized stochastic algorithm (called ADFS) to tackle large-scale finite-sum optimization. The proposed algorithm uses local stochastic proximal updates and randomized pairwise communication between nodes. Some experimental results show the effectiveness of the proposed algorithms. Although the paper is theoretically sound, the current version can be improved in the following aspects: 1. The probability $p_kl$ is a very important parameter, and thus the authors should elaborate on this issue. 2. Does the topology of the network have great influence on the algorithm? If possible, more experiments need to analyze this issue. 3. The two data sets used in the paper are relatively small for distributed stochastic optimization. In particular, it is interested that whether the algorithm is effective for high-dimensional data sets, e.g., RCV1 and KDD2010. 4. Point-SAGA [8] was used to compare with the proposed distributed algorithm, but it is a serial algorithm for stochastic optimization.

Reviewer 3



Motivated by the APCG method for empirical risk minimization, this paper proposed an an accelerated decentralized stochastic algorithm for finite sums. An augmented communication graph is proposed such that the original constrained optimization problem can be transformed to its dual formulation, which can be solved by stochastic coordinate descent. The proposed method employs randomized pairwise communication and stochastic computation. An adaptive sampling scheme for selecting edge is introduced, which is analogous to importance sampling in the literature. The theoretical analysis shows that the proposed algorithm achieves an optimal linear convergence rate for finite-sums, and the time complexity is better than the existing results. The experiments on logistic regression show that it outperforms the related methods and is robust with different network sizes. Overall, I think the theoretical results are novel. It would be better to provide some intuition of the algorithmic design. In particular, the choice of stepsize and R_{k,l} are not well clear. How do we obtain W_{k,l}\Sigma^{-1}y_t in Step 6? Step 9 in Algorithm 1 looks problematic to me. Shouldn't it be v_{t+1}^{i} = z_{t+1}^{(i)} - z_{t+1}^(i, j) + v_{t+1}^{(i, j)}? What is the purpose to have \mu_{i, j}? At the first glance, I thought \mu_{i, j} appears in the communication matrix A, but I get confused when I see (W_k\Sigma^{-1}y_t)^{(k)} \propto \mu_{k,l}^2 in Line 188. The homogenous setting may not be very interesting. One particular application of decentralized computing is mobile computing, in which different users (nodes) have different local sample sizes and networking conditions also vary in different regimes. Thus, I think the paper can be improved if varying local sample size and networking conditions can be considered in the paper. As the proposed ADFS method requires a number of hyper-parameters, how to choose them in practice? I don't find detailed description in either main text or supplementary material. Minor comments: 1. If I understand correctly, the inequality in Line 91 does not hold when batch computations are faster than communications. In this case, "communications are faster than evaluating a single proximal operator ... " would be wrong. 2. There is no definition of Vec(X_{i, j}). 3. The definition of \kappa_{min} in Section D.4 is missing. ******************************************* I have read rebuttal, and my options remain the same. I still encourage the authors to perform experiments on larger datasets (more samples and higher dimensionality).

[Author Response · NeurIPS 2019]

We would like to thank the reviewers for their helpful feedback, including the minor concerns that will be clarified but
which we do not have space to discuss here.

**Significance of the graph reformulation:** The augmented graph allows to cast the stochastic decentralized problem
as a batch decentralized problem with a more complex structure, captured by the matrix $A$. In our case, the augmented
graph contains all the intuition behind the dual formulation (what should the consensus constraints be). More generally,
we believe that it is a powerful way of reasoning to obtain finite sum algorithms from decentralized algorithms that
work with subgraphs of communication. Algorithms and rates can then directly be obtained by studying the Laplacian
of the augmented graph, as done in Appendix D. Yet, it is only needed to obtain the dual formulation.

**Extension of past work:** We mainly refer to the extension of the APCG algorithm to arbitrary sampling and to strong
convexity in subspaces only. This extension is very important for our work because it allows us to choose different
probabilities for communications and for local computations.

**How do we obtain $W_{k\ell}\Sigma^{-1}y_t$:** This comes from the coordinate update $A\nabla_{k\ell}q_A(y_t) = Ae_{k\ell}e_{k\ell}^T A\Sigma^{-1}y_t =$
$\mu_{k\ell}^2 W_{k\ell}\Sigma^{-1}y_t$. The $\mu_{k\ell}$ are indeed related to the communication matrix since the Laplacian matrix of the com-
munication graph is $L = \sum_{k\ell}\mu_{k\ell}^2 W_{k\ell}$. There is a typo and the $\mu_{k\ell}^2$ of line 188 should not appear.

**Hyper parameters:** The only degrees of freedom of ADFS are $p_{k\ell}$ and $\mu_{k\ell}$. The other parameters (such as $R_{k\ell}$, $\rho$, $\eta$,
$\sigma_A$) directly depend on them. For **communication edges**, choosing values of $\mu$ amounts to choosing a gossip matrix,
and choosing $p_{k\ell}$ amounts to tuning how frequently edge $(k, \ell)$ is sampled. Therefore, choosing uniform $p$ and $\mu$ (as in
the experiments) is efficient as long as the graph is not too heterogeneous. For graphs that have non-regular topologies,
choosing $\mu$ better than uniform can be challenging but this is also the case for DSBA and MSDA which consider a
fixed gossip matrix. Besides, MSDA and DSBA are synchronous algorithms that enforce uniform probabilities for
all edges (all edges are activated at each step). Note that we would like to choose $\mu_{k\ell}^2 = p_{k\ell}^2/(R_{k\ell}[\sigma_k^{-1} + \sigma_\ell^{-1}])$ but
this is not possible because $R_{k\ell}$ depends on $\mu_{k\ell}$. For **computation edges**, the theory tells us how to set $p_{k\ell}$ and $\mu_{k\ell}$
using an importance sampling scheme. This only requires knowing $\sigma_k$, the strong convexity of the problem, $L_{k\ell}$, the
smoothness of individual training examples, and $\gamma$, the eigengap of the gossip matrix, which are standard constants
of decentralized optimization problems. In the end, ADFS can be used with parameters given by theory (as given by
Theorem 3), **without any extra tuning**. ADFS may perform better with other choices of $p$ and $\mu$ in some situations but
its main competitors are either forced to use uniform values or require the same tuning.

**Probability $p_{k\ell}$:** The choice of $p_{k\ell}$ impacts both the convergence rate (precisely captured by Inequality (6)) and the
average time per iteration (cruder bound of Theorem 2). Contrary to MSDA, the communication $p_{k\ell}$ can be tuned to
adapt to the topology of the graph, delays and local condition numbers. Although the optimal choice is only clear for
regular graphs, heuristics can be designed using Inequality (6). For example, the communication probabilities can be
chosen as inversely proportional to the edge delay or to the degree of their incident nodes, but this choice is highly
dependent on the setting. Computing probabilities are chosen optimally in the homogeneous setting but could be set
differently as well to reduce waiting time, e.g., to reduce the computational burden of a busy node.

**Datasets used for experiments:** We favored large networks and datasets with many samples over high-dimensional
datasets for our experiments. Yet, we believe that the scale of the experiments is already quite significant. As a
comparison, experiments in the MSDA paper use synthetic datasets of dimension 10 and experiments in the DSBA
paper use a network of size 10. We use 100 computing nodes and $10^6$ samples in total, which is significantly larger.
Finally, serial APCG performs well on RCV1 and ADFS is a decentralized version of APCG, so we expect it to have
good performances on RCV1 as well. We will add a full set of experiments on RCV1 to a revised version of the paper.

**Comparison with Point-SAGA:** As written in the introduction, Point-SAGA is a serial algorithm indeed and we
apologize if this was not clear enough. Yet, we thought that it was interesting to have it as a competitor because it beats
many distributed algorithms when the number of machines is not too big, as shown in Figure 3 (a). Our point was that
ADFS compares nicely with optimal single machine algorithms on one machine (which is not the case of many other
algorithms) while being able to take advantage of using more machines (Point-SAGA is only designed for one machine,
as Reviewer 2 pointed out). We will make the distinction clearer by adding "serial" to the legend.

(a) ADFS on several graphs

**Influence of the network topology:** As long as all nodes have roughly the same probability of starting an update (to avoid waiting times), the topology of the network mainly influences the iteration complexity of ADFS through the constant $\gamma$. Figure 1a shows experiments with 100 nodes and 500 samples per node on the covtype dataset. For the geographic graph, each node is randomly placed in the unit square and nodes that are at distance less than $r$ are neighbours. As expected, the convergence rate improves with $\gamma$. Yet, ADFS is slower on $Geo(0.17)$ than on the grid because nodes have uneven degrees so waiting time increases (because all edges have equal probabilities). A non-uniform probability distribution could speed-up convergence for $Geo(0.17)$.

[Meta-Review · NeurIPS 2019]

This paper addresses optimization under communication constraints via decentralized proximal algorithms. Optimization under communication constraints is an important area of machine learning and there is still much to be gained from rigorous research in this area. Statistical literature does not concern itself much with inter- and intra-processor communication; nor does optimization (operation research) literature. Machine learning is a natural community for this research to be advanced. This paper uses a randomization scheme for communicating information between nodes to achieve impressive learning rates. The authors derive theoretical bounds on the estimation error induced by the communication constraint and show that they achieve improvements over state-of-the-art approaches with some empirical experiments. The results seem to be technically sound and significant. The reviewers noted value in estimates of synchronization times and the combination of acceleration and asynchronous aspects. One reviewer noted an important dependency on a parameter in the analysis, and the authors took steps to address that aspect in their response. However, that review remained unchanged after the author response and was not consistent with the other two reviews. While the main contributions of this paper and theoretical, not empirical the theoretical ideas are important and, in my view, will be important for driving research in this field forward. I strongly recommend this paper for acceptance.